# Risk-stratification of febrile African children at risk of sepsis using sTREM-1 as basis for a rapid triage test

Aleksandra Leligdowicz [1], Andrea L. Conroy [2], Michael Hawkes[3], Melissa Richard-Greenblatt[4], Kathleen Zhong[4], Robert O. Opoka[5], Sophie Namasopo[6], David Bell [7], W. Conrad Liles[8], Bruno R. da Costa[9], Peter Jüni[9,11] & Kevin C. Kain [10,11✉]

Identifying febrile children at risk of sepsis in low-resource settings can improve survival, but recognition triage tools are lacking. Here we test the hypothesis that measuring circulating markers of immune and endothelial activation may identify children with sepsis at risk of all-cause mortality. In a prospective cohort study of 2,502 children in Uganda, we show that Soluble Triggering Receptor Expressed on Myeloid cells-1 (sTREM-1) measured at first clinical presentation, had high predictive accuracy for subsequent in-hospital mortality. sTREM-1 had the best performance, versus 10 other markers, with an AUROC for discriminating children at risk of death of 0.893 in derivation (95% CI 0.843–0.944) and 0.901 in validation (95% CI 0.856–0.947) cohort. sTREM-1 cutoffs corresponding to a negative likelihood ratio (LR) of 0.10 and a positive LR of 10 classified children into low (1,306 children, 53.1%), intermediate (942, 38.3%) and high (212, 8.6%) risk zones. The estimated incidence of death was 0.5%, 3.9%, and 31.8%, respectively, suggesting sTREM-1 could be used to risk-stratify febrile children. These findings do not attempt to derive a risk prediction model, but rather define sTREM-1 cutoffs as the basis for rapid triage test for all cause fever syndromes in children in low-resource settings.

[1] Robarts Research Institute, University of Western Ontario, 1151 Richmond St, London, ON N6A 3K7, Canada. [2] Department of Pediatrics, Indiana University School of Medicine, 1044 West Walnut St., Building 4, Indianapolis, IN 46202, USA. [3] Division of Pediatric Infectious Diseases, 3-593 Edmonton Clinic Health Academy, University of Alberta, Edmonton, AB T6G1C9, Canada. [4] Toronto General Hospital, University Health Network, Sandra Rotman Centre for Global Health, MaRS Centre, 101 College St. TMDT 10-360A, Toronto, ON M5G 1L7, Canada. [5] Department of Paediatrics and Child Health, Mulago Hospital and Makerere University, Kampala, Uganda. [6] Department of Pediatrics, Kabale District Hospital, Kabale, Uganda. [7] Independent consultant, Issaquah, WA 98027, USA. [8] Departments of Medicine, Pathology, Global Health, and Pharmacology, 1959 NE Pacific Street; HSB RR-511, Box 356420, University of Washington, Seattle, WA 98195-6420, USA. [9] Applied Health Research Centre, Li Ka Shing Knowledge Institute of St Michael's Hospital, Institute of Health Policy, Management and Evaluation, University of Toronto, 30 Bond St, Toronto, ON M5B 1W8, Canada. [10] Tropical Disease Unit, Sandra Rotman Centre for Global Health, Toronto General Hospital, University Health Network, Department of Medicine, University of Toronto, MaRS Centre, 101 College St. TMDT 10-360A, Toronto, ON M5G 1L7, Canada. [11] These authors contributed equally: Peter Jüni, Kevin C. Kain. ✉email: kevin.kain@uhn.ca

Sepsis is defined as life-threatening organ dysfunction caused by a dysregulated host response to infection and is a leading cause of death in children under 5 in low-and-middle-income countries. In 2017 there were an estimated 13 million cases of sepsis and 2.5 million sepsis-related deaths in sub-Saharan Africa with 25% of pediatric sepsis cases attributed to malaria[1–3].

Sepsis is treatable and the early identification of febrile children at risk of sepsis can improve survival[1,2]. However, effective tools for their prompt and accurate recognition at the community level are lacking, and ~50% of deaths occur at home[3].

Immune and endothelial activation are implicated in the pathogenesis of sepsis, including severe malaria[4–13]. Measuring circulating mediators of these pathways at first clinical presentation could identify children with impending sepsis, enabling early recognition and triage. We tested this hypothesis in a prospective cohort of febrile children presenting to the emergency department of a regional hospital in Uganda to determine if these plasma markers can risk-stratify children with fever due to *Plasmodium falciparum* malaria and non-malarial causes. We compared the performance of 11 plasma immune and endothelial activation mediators to identify the marker with the highest predictive accuracy for predicting 7-day mortality. We validated the performance of the top biomarker using a two-step validation process to confirm robustness of our findings. Our goal was to identify a mediator with a pathobiologic link to sepsis that could enable the development of a rapid triage test to predict mortality from any cause in febrile children at the community level in low-resource settings.

## Results

**Cohort characteristics**. Between February 15, 2012 and August 29, 2013, we consecutively enrolled 2,502 febrile children, with 1,433 children up to Oct 31, 2012 included in the derivation cohort, and 1,069 children from Nov 1, 2012 onwards included in the validation cohort (6). During the period of interest of 7 days, 2,039 children were regularly discharged or survived up to 7 days (81.5%), 95 children had died (3.8%), 337 absconded (13.5%) and 31 were transferred (1.2%). Table 1 shows a comparison of baseline characteristics of the 95 children who died up to 7 days with the 2,407 children who survived until regular discharge from hospital, abscondment or transfer. Children who died within 7 days had a greater severity of illness (higher LODS score, higher

lactate levels) and were less likely to be *P. falciparum* malaria positive. Supplementary Table 1 presents this comparison separately for derivation and validation cohorts. All children were evaluated promptly (Table 1) and treated according to national guidelines (Supplementary Table 2).

Supplementary Table 3 compares baseline characteristics between derivation and validation cohorts. Even though the characteristics of children between the derivation and validation cohort appeared similar, there were 43 deaths up to 7 days in the derivation cohort (3.0%) and 52 in the validation cohort (4.9%). After multiple imputation, the estimated incidence of death up to 7 days was 3.9% (95% CI 2.8 to 4.9%) in the derivation cohort and 5.5% (95% CI 4.1 to 7.0%) in the validation cohort (odds ratio 1.45, 95% CI 0.98 to 2.15). Supplementary Tables 4 and 5 present comparisons of baseline characteristics between the 2039 children who were regularly discharged from hospital or survived up to 7 days, the 337 children who absconded, and the 31 children who were transferred up to 7 days. Forty-two children were excluded from all analyses of biomarkers as no plasma sample was available (1.7%). An additional 376 children were excluded from comparative performance analyses of the 11 biomarkers (15.0%) as they had absconded or were transferred before or after 7 days (Fig. 1).

**Comparative performance of 11 markers of immune and endothelial activation**. A total of 2,084 children were included in comparative performance analyses of biomarkers, with 1,176 children analysed in the derivation cohort, and 908 in the validation cohort (Fig. 1). Table 2 and Supplementary Fig. 1a-b show AUROCs for the discrimination between children who died up to 7 days and children who survived in derivation, primary validation (bootstrapping in derivation cohort), and secondary validation. As previously described in febrile adults[14], sTREM-1 showed the best discrimination, with an AUROC of 0.893 in derivation (95% CI 0.843 to 0.944), 0.894 in primary validation (95% CI 0.844 to 0.944) and 0.901 in secondary validation (95% CI 0.856 to 0.947). The AUROC of 0.859 of the second ranked biomarker, soluble fms-like tyrosine kinase 1 (sFlt-1), appeared optimistic in derivation but decreased by 0.063 in secondary validation. Supplementary Table 6 presents geometric means with 95% reference ranges in children who survived or died for the combined cohorts included in the comparative performance analysis, and separately for derivation and validation cohorts. Supplementary Table 7 presents AUROCs

**Table 1 Baseline characteristics of children who died within 7 days and children who survived until regular discharge from hospital, abscondment or transfer.**

| Characteristic at baseline | Dead (*n* = 95) | Alive (*n* = 2407) | Odds ratio (95% CI) | *P*-value |
|---|---|---|---|---|
| Age, months | 18.2 (12.9) | 19.7 (12.9) | 0.78 (0.51 to 1.21) | 0.27 |
| Male (*n*, %) | 54 (56.8) | 1321 (54.9) | 1.06 (0.70 to 1.61) | 0.77 |
| Time to MD, h | 1.6 (1.7) | 3.0 (2.4) | 0.17 (0.09 to 0.33) | <0.001 |
| Temperature | 37.3 (1.3) | 37.9 (1.2) | 0.40 (0.26 to 0.61) | <0.001 |
| SpO$_2$% | 90.7 (11.3) | 97.1 (4.0) | 0.35 (0.27 to 0.46) | <0.001 |
| Heart rate | 160.4 (32.8) | 160.6 (24.1) | 0.98 (0.65 to 1.50) | 0.94 |
| LODS (*n*, %) | | | | <0.001 |
| 0 | 3 (3.2) | 1512 (62.8) | 1.00 (reference) | |
| 1 | 7 (7.4) | 437 (18.2) | 8.07 (2.07 to 31.37) | |
| 2 | 25 (26.3) | 287 (11.9) | 43.96 (13.17 to 146.79) | |
| 3 | 60 (63.2) | 172 (7.1) | 176.29 (54.62 to 568.95) | |
| Lactate, mmol/L | 7.4 (1.3 to 41.4) | 3.3 (0.8 to 14.0) | 6.86 (4.56 to 10.32) | <0.001 |
| *P. falciparum* malaria (n, %) | 37 (38.9) | 1292 (53.7) | 0.55 (0.36 to 0.84) | 0.005 |
| HIV (*n*, %) | 5 (5.3) | 45 (1.9) | 2.97 (1.08 to 8.22) | 0.036 |

Odds ratios, 95% confidence intervals and *P*-values are from univariate logistic regression models.
*CI* confidence interval.

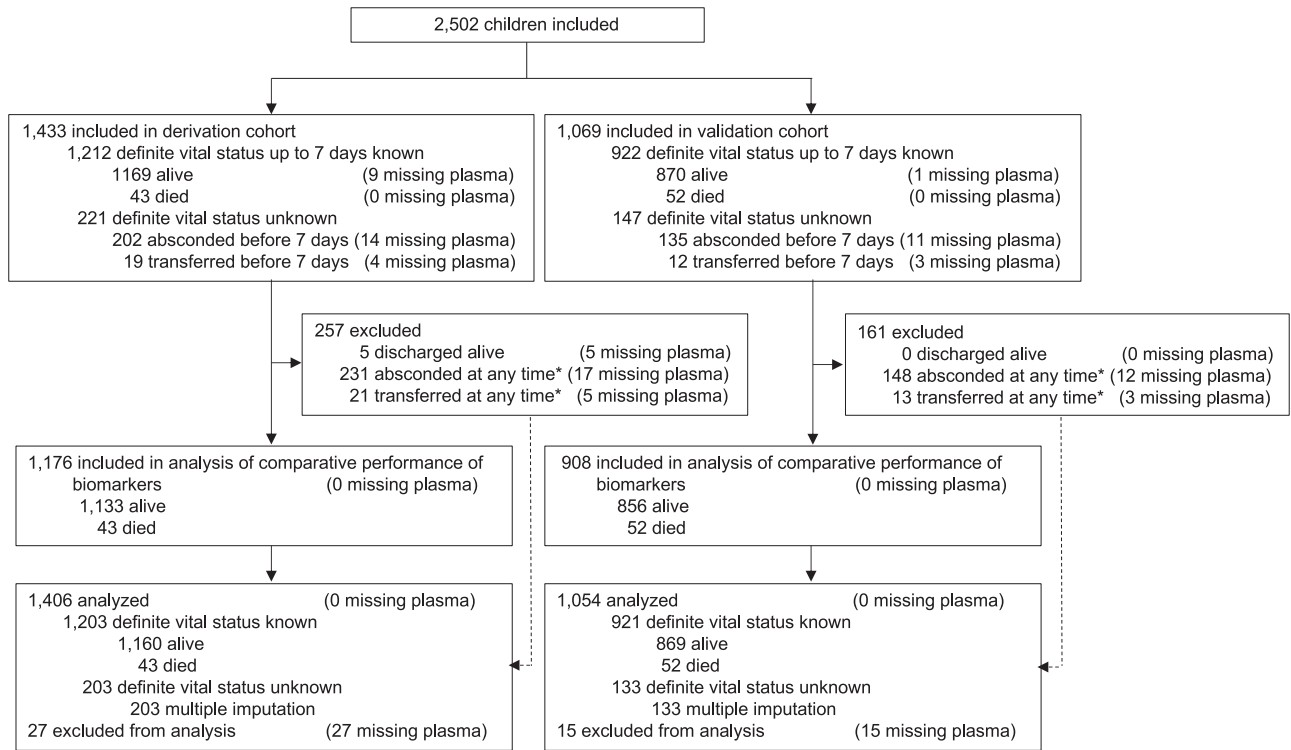

**Fig. 1 Flow of consecutively enrolled children in the prospective cohort and included in analysis.** In the derivation cohort, 29 children absconded after 7 days and 2 children were transferred after 7 days; in the validation cohort, 13 children absconded after 7 days and 1 child was transferred after 7 days; by definition, the vital status up to 7 days was known for these children, but they were excluded from the analysis of comparative performance of biomarkers.

**Table 2 Comparative performance of biomarkers for children who died up to 7 days or who survived in derivation, primary and secondary validation cohort.**

| Biomarker | Derivation (n = 1,176) | | Primary validation (n = 1,176) | | Secondary validation (n = 908) | |
|---|---|---|---|---|---|---|
| | AUROC (95% CI) | P-value | AUROC (95% CI) | P-value | AUROC (95% CI) | P-value |
| sTREM-1 | 0.893 (0.843 to 0.944) | – | 0.894 (0.844 to 0.944) | – | 0.901 (0.856 to 0.947) | – |
| sFlt1 | 0.859 (0.792 to 0.926) | 0.103 | 0.860 (0.792 to 0.927) | 0.102 | 0.796 (0.728 to 0.864) | ≤0.001 |
| IL-8 | 0.843 (0.772 to 0.914) | 0.105 | 0.845 (0.775 to 0.916) | 0.115 | 0.790 (0.712 to 0.868) | ≤0.001 |
| Ang-2 | 0.846 (0.787 to 0.905) | 0.081 | 0.847 (0.789 to 0.904) | 0.084 | 0.784 (0.715 to 0.853) | ≤0.001 |
| CHI3L1 | 0.826 (0.750 to 0.902) | 0.054 | 0.829 (0.754 to 0.904) | 0.064 | 0.771 (0.702 to 0.840) | ≤0.001 |
| sTNFR1 | 0.783 (0.696 to 0.870) | ≤0.001 | 0.785 (0.702 to 0.868) | ≤0.001 | 0.803 (0.726 to 0.879) | 0.002 |
| IL-6 | 0.821 (0.743 to 0.900) | 0.064 | 0.824 (0.749 to 0.899) | 0.065 | 0.753 (0.673 to 0.832) | ≤0.001 |
| sICAM-1 | 0.663 (0.561 to 0.764) | ≤0.001 | 0.666 (0.566 to 0.765) | ≤0.001 | 0.620 (0.535 to 0.704) | ≤0.001 |
| sVCAM-1 | 0.660 (0.561 to 0.759) | ≤0.001 | 0.662 (0.560 to 0.763) | ≤0.001 | 0.608 (0.534 to 0.682) | ≤0.001 |
| IP-10 | 0.581 (0.486 to 0.677) | ≤0.001 | 0.583 (0.489 to 0.677) | ≤0.001 | 0.566 (0.484 to 0.648) | ≤0.001 |
| Ang-1 | 0.347 (0.261 to 0.433) | ≤0.001 | 0.350 (0.261 to 0.439) | ≤0.001 | 0.331 (0.262 to 0.401) | ≤0.001 |

*P*-values correspond to difference in AUROC as compared to AUROC of sTREM-1 and were derived from Chi-squared tests. Primary validation was based on 500 bootstrap samples with replacement in the derivation cohort. Secondary validation was performed in the temporally defined validation cohort.
*AUROC* area under the receiver operating characteristic curve, *CI* confidence interval.

of the 4 biomarkers that were quantified in all children with available plasma. sTREM-1 showed again the best discrimination, with an AUROC of 0.875 in derivation (95% CI 0.826 to 0.924), 0.876 in the primary validation (95% CI 0.825 to 0.928) and 0.885 in the secondary validation (95% CI 0.841 to 0.929). Supplementary Table 8 presents AUROCs separately for children with and without diagnosis of *P. falciparum* malaria in derivation, primary and secondary validation; results were similar. The updated AUROC for sTREM-1 based on the pooled data of all 2,460 children of the derivation and validation cohorts combined was 0.879 overall (95% CI 0.847 to 0.912), 0.931 in children with *P. falciparum* malaria

(95% CI 0.910 to 0.951) and 0.871 in children without malaria (95% CI 0.825 to 0.917).

sTREM-1 predictive performance of 7-day mortality relative to the best clinical score (LODS) was statistically similar in the derivation (AUROC for sTREM-1 0.894 (95% CI 0.843 to 0.944) versus for LODS 0.907 (95% CI 0.869 to 0.944), *P* = 0.661) and in the validation (AUROC for sTREM-1 0.901 (95% CI 0.856 to 0.946) versus for LODS 0.912 (95% CI 0.875 to 0.949; *P* = 0.628) cohort.

In contrast, sTREM-1 predictive performance of 7-day mortality relative to C-reactive protein (CRP) and procalcitonin

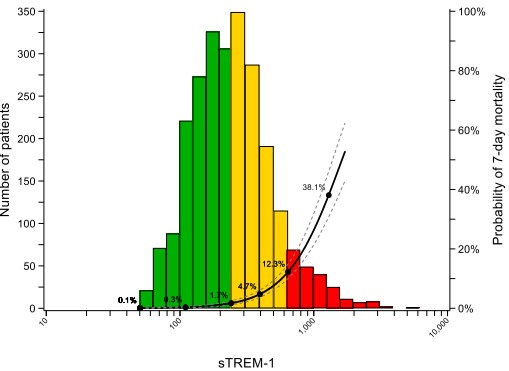

**Fig. 2 Distribution of sTREM-1 at presentation with predicted probability of 7-day mortality in the combined cohort.** Histogram refers sTREM-1 distribution. Negative and positive Likelihood Ratios (LRs) in the derivation and validation cohorts combined were used to risk-stratify febrile children: "green" zone: low risk (LR− of 0.10, sTREM-1 < 239 pg/mL), "yellow" zone: refer and monitor (sTREM-1 ≥ 239 pg/mL and <629 pg/mL), "red" zone: urgent admission/support (LR+ of 10, sTREM-1 ≥ 629 pg/mL).

(PCT) was significantly greater in the derivation (AUROC for sTREM-1 0.872 (95% CI 0.812 to 0.932) versus for CRP 0.645 (95% CI 0.547 to 0.742), $P < 0.0001$ and for PCT 0.667 (95% CI 0.565 to 0.769), $P < 0.0001$) and in the validation (AUROC for sTREM-1 0.879 (95% CI 0.823 to 0.936) versus for CRP 0.597 (95% CI 0.501 to 0.693), $P < 0.0001$ and for PCT 0.695 (95% CI 0.605 to 0.785), $P < 0.0001$) cohort.

**Risk stratification based on sTREM-1 levels.** A total of 2,460 children were included in analyses, with 1,406 children analysed in the derivation cohort, and 1,054 in the validation cohort. The cut-offs based on the pooled data of the 2,460 children in derivation and validation cohorts combined were 239 pg/mL and 629 pg/mL. Supplementary Table 9 presents likelihood ratios and Supplementary Table 10 presents false positive and false negative rates according to fixed cut-offs of sTREM-1, which were similar between derivation, primary and secondary validation for all cut-offs.

Figure 2 presents the distribution of sTREM-1 levels and corresponding probabilities of death up to 7 days predicted from logistic regression. Out of 2,460 children with available plasma, 1,306 children had sTREM-1 levels of less than 239 pg/mL and were classified in the green low risk zone (53.1%), 942 children had levels of 239–629 pg/mL and were classified in the yellow intermediate risk zone (38.3%), whereas 212 children had levels above 629 pg/mL and were classified in the red high risk zone (8.6%), with deaths observed in 3 (0.2%), 30 (3.2%) and 62 children (29.3%), respectively. After multiple imputation, accounting for missing vital status in children who absconded or were transferred, the estimated incidence of death in the derivation and validation cohorts combined was 0.5%, 3.9% and 31.8% in green, yellow and red zones, respectively.

The distribution of sTREM-1 levels and corresponding probabilities of death are presented in the Supplementary Information separately for derivation (Supplementary Fig. 2a) and validation (Supplementary Fig. 2b) cohorts. The estimated incidence of death in green, yellow and red zones was 0.3% (95% CI 0.0 to 0.7%), 3.3% (95% CI 1.7 to 4.8%) and 26.5% (95% CI 18.5 to 34.6%) in the derivation cohort, and 0.8% (95% CI 0.0 to 1.6%), 4.8% (95% CI 2.6 to 7.1%) and 38.8% (95% CI 28.6 to 49.1%) in the validation cohort. Calibration plots from primary and secondary validation showed adequate calibration for green and yellow zones in both primary and secondary validation, adequate calibration for the red zone in primary validation, but

higher mortality than predicted for the red zone in secondary validation (Supplementary Fig. 3 and 4). Accordingly, the calibration-in-the-large was −0.064 in primary validation, but 0.514 in secondary validation (Supplementary Fig. 4). Figure 3 presents time-to-event analyses in the overall population (top), in the subgroup of children with diagnosis of *P. falciparum* malaria (middle) and without diagnosis of malaria (bottom). In the derivation and validation cohorts combined, children in the green zone had an estimated incidence of death of 0.5% (95% CI 0.1 to 0.9%), children in the yellow zone 3.9% (95% CI 2.6 to 5.2%) and children in the red zone 31.8% (95% CI 25.4 to 38.2%; omnibus *p*-value for difference between 3 zones <0.0001). In children with diagnosis of *P. falciparum* malaria, those in the green zone had an estimated incidence of death of 0.0% (95% CI 0.0 to 0.5%), children in the yellow zone 1.6% (95% CI 0.5 to 2.8%), and children in the red zone 23.6% (95% CI 16.3 to 30.9%; omnibus *p*-value for difference between 3 zones 0.003). In children without diagnosis of *P. falciparum* malaria, those in the green zone had an estimated incidence of death of 0.9% (95% CI 0.1 to 1.6%), children in the yellow zone 7.4% (95% CI 4.7 to 10.2%), and children in the red zone 45.9% (95% CI 34.5 to 57.3%; omnibus *p*-value for difference between 3 zones <0.0001). Supplementary Fig. 5 and 6 show corresponding time-to-event analyses in derivation (Supplementary Fig. 5) and validation (Supplementary Fig. 6) cohorts separately, which showed similar results.

## Discussion

In this study, sTREM-1, a cell surface receptor expressed on myeloid cells associated with neutrophil and monocyte response amplification[15], was superior to ten other biomarkers of endothelial or immune activation in predicting mortality in febrile children aged 2 months to 5 years presenting to the emergency department of a regional hospital in Uganda. The AUROC was 0.893 in the derivation cohort and 0.901 in the validation cohort, very similar to the previously reported AUROC of 0.87 in predicting death within 28 days in 507 consecutive febrile adults presenting to four outpatient clinics in Tanzania[14]. Similar to the adult study, discrimination was independent of aetiology and comparable in children with and without malaria. Taken together, this suggests that multiple life-threatening infections share common pathways of injury that are independent of pathogen, age and geographic location[6–8,12,13,16]. We derived and validated cut-offs of sTREM-1 as the basis for the development of a rapid test (i.e.: lateral flow) that could be used as a simple triage tool at the community level in low-resource settings. Cut-offs associated with LR− of 0.10 and LR+ of 10 allowed us to classify more than half of the children into a green zone considered to be at low risk of death of less than 1%, whereas less than 10% of children were classified into a red zone considered to be at high risk of death of 25% or more. Approximately 40% remained in an intermediate risk category, with an estimated risk of death of 3–5%.

The majority of paediatric infections are self-limited, and few are life threatening. In the absence of critical illness, many febrile syndromes can be treated conservatively[17]. However, we currently lack effective tools to identify children at risk of progression to severe illness—a priority that is not addressed by pathogen-based diagnostics. This results in increased mortality in those with life-threatening infections[18], while paradoxically causing harm[19], misallocation of scarce resources related to over-admission and antimicrobial treatment and added risk of nosocomial infection in children with milder self-limited infections[20].

Our results suggest that a rapid triage test based on finger-prick blood sample using sTREM-1 as a disease severity marker could be used as a simple, objective and clinically meaningful risk-stratification tool that could facilitate an integrated approach to

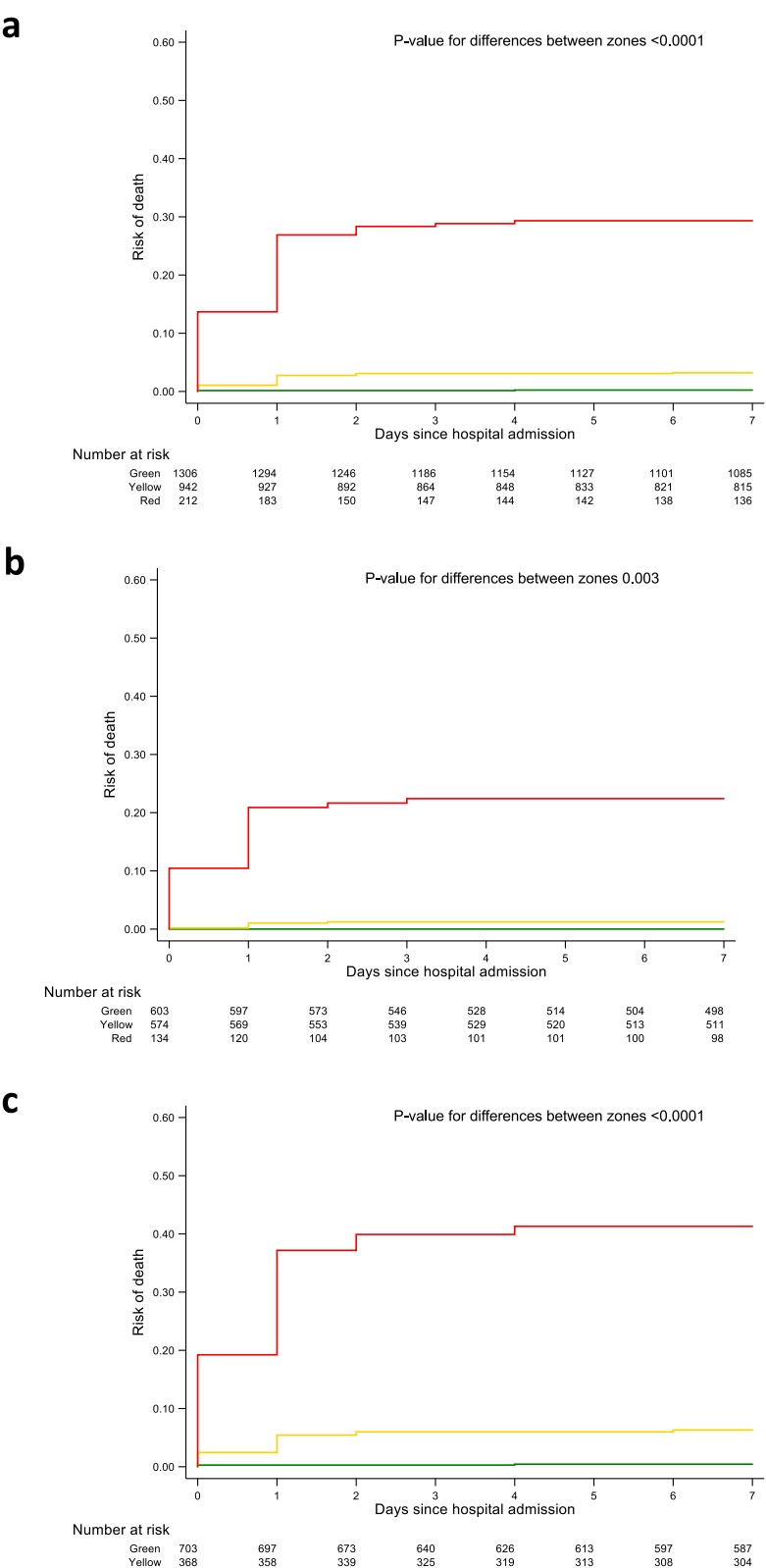

**Fig. 3 Time-to-event analyses.** Cumulative incidence of hospital death in the combined cohort (**a**), in the subgroup of children with *P. falciparum* malaria (**b**) and without malaria (**c**), stratified into the "green", "yellow", "red" sTREM-1 zones. sTREM-1 cut-off values were generated using a LR− of 0.10 (<239 pg/mL) and LR+ of 10 (≥629 pg/mL) derived in the derivation and validation cohorts combined corresponding to the mortality risk zones. Time-to-event curves were plotted based on Kaplan–Meier estimates. Wald test was used to calculate *P*-values for differences between zones.

manage fever syndromes at the community level, where no medically qualified health professionals may be available to triage children based on clinical criteria. Children in the red zone according to the rapid triage test, who are at high risk of death, could be urgently referred and prioritized for hospital care, children in the yellow intermediate risk zone could be referred with lower priority for monitoring, diagnostic workup and management depending on the clinical course, whereas children in the green zone could be considered for management at the community level. This strategy could decrease the referral of children with uncomplicated or self-limited infections who are unlikely to benefit from admission, investigation, and urgent supportive care. Collectively this could result in task-shifting from scarce, highly trained healthcare professionals in centralized health units to community health workers, decrease the pressure on healthcare facilities and professionals, enhance appropriate resource allocation and potentially decrease sepsis-related mortality in children. The strategy of using a rapid point of care test such as a lateral flow test to triage febrile children at the community level has multiple attributes, including ease-of-use, speed, cost, cultural acceptability, gender equity in access to care, and evidence-based decision-making, that would support their implementation and scalability.

Our study has several limitations. First, this is a single centre study and our results will need to be replicated by independent groups in different settings. However, our results are very similar to those derived in consecutive febrile adults presenting for out-patient care in Tanzania[14], supporting the generalizability of our findings. Second, our study was complicated, as is common in low-resource settings[21], by children who absconded or were transferred, for whom vital status could not be definitely ascertained. Plasma was also missing, but in less than 2% of children. We used multiple imputation to account for missing vital status and consider the missing at random assumption of the multiple imputation model given the observed data plausible[22]. Third, calibration was only modest in the secondary validation, as the predicted mortality for children in the red zone was lower than observed in this cohort, which likely reflects the trend towards higher mortality in the validation cohort compared to the derivation cohort. However, this does not alter the suggested strategy for triage: children in the red zone would be at high risk of death, regardless of the actual risk of 1 in 4 in the derivation cohort, or 1 in 3 in the validation cohort. Fourth, our study did not attempt to derive and validate a risk prediction model but rather to derive and validate the use of suitable sTREM-1 cut-offs as the basis for a rapid test for prospective risk stratification of febrile children. However, the study was conducted in a single prospective cohort study and randomized trials will be required to confirm these cut-offs and to establish that the addition of a rapid triage test to current standard of care at the community level will improve outcomes of febrile children in low-resource settings. Lastly, other acute phase proteins, such as ferritin, are increased during infection. Although this marker is associated with severity of illness, it is not specific to the pathobiology of febrile illness and as such, was not included in this study. Of note, the predictive performance of 7-day mortality of CRP and PCT, well-studied non-specific markers of inflammation, was significantly lower for relative to sTREM-1.

Strengths of our study include its prospective design that distinguish it from prior retrospective cohorts studies that modelled mortality in children in resource-limited African settings[6,23], direct comparison of multiple candidate markers of disease severity at the earliest time point of healthcare presentation, the large sample size with a sufficient number of outcome events, the confirmatory nature of our results, with a discrimination nearly identical to what was previously reported in febrile adults in Tanzania[14] as well as other smaller studies conducted in children (Supplementary Table 11)[6,24–37] and adults[38,39], the robustness of results in primary and secondary validation, and the biological plausibility of the observed association[15].

In conclusion, sTREM-1, a severity marker with a pathophysiologic link to sepsis, measured at clinical presentation, accurately predicted mortality in febrile children with either *P. falciparum* malaria or non-malarial aetiology, in a regional hospital in Uganda. Simple risk stratification based on a rapid sTREM-1 test could enhance triage and improve outcomes in resource-limited settings.

## Methods

**Design and population**. This was a prospective cohort study in children aged 2 months to 5 years presenting to the emergency department with a history of fever in the past 48 h or an axillary temperature >37.5 °C, and admitted to the Jinja Regional Hospital in Uganda between February 15, 2012 and August 29, 2013 according to the treating physician's judgement[40]. Children enrolled up to Oct 31, 2012 were prospectively considered as part of the derivation cohort and children included after this date as part of the validation cohort. The hospital serves a catchment area of three million people from 12 districts in mid-eastern Uganda. A plasma sample was collected at the time of emergency room presentation prior to initiation of treatment. Patients were managed according to national algorithms for the treatment of malaria, pneumonia, respiratory distress, anaemia, and hypoglycaemia (see Supplementary Information). Data were collected on paper Clinical Report Forms (CRF) and manually transcribed into Microsoft Excel (version 11.0).

The study was approved by the Uganda National Council for Science and Technology, Makerere University Research Ethics Committee (Kampala, Uganda, REC Protocol # REF 2011-255), the University Health Network (Toronto, Canada, REB number 12-0039-AE), and was registered on clinicaltrials.gov (identifier: NCT04726826). The parent or caregiver of every study participant provided written informed consent. The numbers of children screened and the number of eligible children whose parents or caregivers refused consent were recorded in a screening log, which was lost after completion of recruitment. Therefore, the exact number of eligible children whose parents or caregivers refused consent is unknown but was estimated by local research staff to be 25 or less.

**Quantification of biomarkers**. Eleven of 11 biomarkers of endothelial or immune activation[6–13] used in our previously reported study in febrile adults[14] were quantified using the multiplex Luminex® platform (Luminex, Austin TX) with custom-developed reagents (R&D Systems, Minneapolis, MN)[41] in the 2,084 children with available plasma who had been regularly discharged from hospital or had died (see Supplementary Information). sTREM-1 and sFlt-1, which ranked first and second in the analysis of the derivation cohort, sTNFR1, which ranked second after sTREM-1 in the previously reported study in febrile adults in Tanzania[14], and Ang-2, which was the biomarker used for sample size considerations of the current study, were subsequently quantified using identical methods in the remaining 376 children with available plasma who were transferred or absconded. All samples were processed and analysed blinded to clinical outcome. Eight samples on each plate were performed in duplicate to ensure intra-assay consistency (coefficients of variance shown in Supplementary Table 12).

**Statistical analysis**. The sample size consideration is described in the Supplementary Information. It assumed a nested case-control design, defining in-hospital deaths as cases and using 3 survivors per case as matching controls. The current analysis reflects the original prospective cohort design rather than a nested case-control design. For the analysis of the comparative performance of all 11 biomarkers, we used a non-parametric approach to determine the AUROC as a measure of discrimination between children who died from any cause up to 7 days and children who survived, and ranked biomarkers according to the estimated AUROC in children of the derivation cohort with an available plasma sample who had died or were regularly discharged from hospital (n = 1,176), but were neither transferred to another hospital nor absconded[42]. Next, we ranked the 4 biomarkers according to AUROCs estimated in the complete derivation cohort of children with an available plasma sample (n = 1,406). Since the predictive value of sTREM-1 previously reported in febrile adults[14] was confirmed in all analyses of the derivation cohort, we identified the sTREM-1 levels in pg/mL that were associated with a negative likelihood ratio (LR−) of 0.10 and a positive likelihood ratio (LR+) of 10. A LR− of 0.10 indicates that it was 10 times less likely to find a sTREM-1 concentration at clinical presentation, lower than the associated cut-off in children who subsequently died as compared with those who survived. A LR+ of 10 indicates that it was 10 times more likely to find a sTREM-1 concentration equal to or higher than the associated cut-off in children who subsequently died as compared with those who survived[43]. The pre-specified targets of 0.10 for the LR− and 10 for the LR+ are considered to be associated with large, often conclusive changes from pre-test to post-test probabilities[44]. Likelihood ratios were preferred over predicted risks to derive cut-offs for the projected rapid triage test, as they did not

depend on variations in the underlying risk of death in the studied population. Children with sTREM-1 values below the cut-off associated with a LR− of 0.10 were classified as low risk (green zone), children with sTREM-1 values above the cut-off associated with a LR+ of 10 as high risk (red zone). Remaining children were classified as intermediate risk (yellow zone). For descriptive purposes, we also estimated LR− and LR+ for different cut-offs of sTREM-1 (see Supplementary Information). Using logistic regression, we estimated risks of in-hospital death up to 7 days in green, yellow and red zones, and predicted the association between log sTREM-1 levels and the logit of mortality up to the 99th percentile of the distribution of sTREM-1 levels, back transformed logits to probabilities and superimposed them onto the corresponding distribution of sTREM-1 levels on a logarithmic scale. Discrimination based on AUROCs, associated rankings of biomarkers, likelihood ratios associated with identified cut-offs of sTREM-1 in the derivation cohort, calibration of mortality risks in green, yellow and red zones, and calibration of mortality risks predicted from log sTREM-1 levels were validated in the temporally defined derivation cohort based on 500 bootstrap samples with replacement (primary validation)[45], and then validated in the temporally defined validation cohort (secondary validation). Calibration was defined as the agreement between observed and predicted mortality risk and assessed in calibration plots and calibration-in-the-large[45]. Because primary and secondary validations were successful, we updated the model based on the pooled data of the all children with available plasma of temporally defined derivation and validation cohorts combined ($n = 2,460$) to make full use of all available information when determining AUROCs, sTREM-1 cut-offs associated with a LR− of 0.10 and a LR+ of 10, and estimated mortality risks. Finally, we plotted time-to-event curves based on Kaplan–Meier estimates to compare in-hospital mortality from any cause up to 7 days between green, yellow and red zones in the overall population and in subgroups with and without diagnosis of malaria. Analyses of the comparative performance of the 11 biomarkers were based on children who did not abscond, were not transferred and did not have a missing plasma sample ($n = 2,084$). Remaining biomarker analyses were based on all children with an available plasma sample ($n = 2,460$); children without a plasma sample ($n = 42$) were excluded throughout. To account for missing vital status in children who were transferred or absconded before 7 days we used multiple imputation (see Supplementary Information)[22], with all baseline characteristics and in-hospital death up to 7 days as variables in the imputation model to create 20 imputed datasets. When deriving time-to-event curves for descriptive purposes, however, we censored children who were transferred to another hospital or absconded before 7 days at the day of transfer or abscondment. Statistical analyses were performed using Stata 15.1 (StataCorp, College Station, TX).

## Supplementary methods

*Sample size consideration.* The original sample size consideration assumed a nested case-control design, defining in-hospital deaths as cases and using 3 survivors per case as matching controls. The recruitment of 2,475 children would result in 99 in-hospital deaths and more than 80% power to detect a standardized difference between cases and controls of 0.45 standard deviations using a Bonferroni correction that allowed this difference to be tested simultaneously for 13 biomarkers at a two-sided alpha of 0.0038 (0.05/13). The difference of 0.45 standard deviation units would correspond to a 40% difference in mean values of Ang-2, between a mean of 11,000 pg/mL in survivors and 15,600 pg/mL in children who died in-hospital, with a common standard deviation of 9,600 pg/mL[6]. The current analysis reflects the original prospective cohort design rather than a nested case-control design, using in-hospital mortality from any cause up to 7 days as pre-specified primary outcome.

*Patient enrolment.* Patient enrolment occurred between 08:00 and 20:00. Patients presenting after 20:00 had a sample collected on the next day. Of 2,502 enrolled children, 11 presented after working hours, 8 of whom had follow-up to hospital discharge and were included in the analysis. The median time from presentation to sample collection in children who were included in analysis was 2.6 h (interquartile range (IQR) 1.0, 4.1).

*Blood sample processing.* A venipuncture blood sample was collected at the time of presentation to the emergency room into a BD Microtainer® blood collection tube with EDTA (Becton, Dickinson and Company) and processed within 4 h. Microtainer tubes were centrifuged for 20 min at $1,360 \times g$ and the plasma layer was aspirated and stored in a cryovial at +4 °C until transport from Jinja, Uganda to Kampala, Uganda. Sample transport to Kampala, Uganda was done on ice and occurred once a day. In Kampala, cryovials were stored in a −80 °C freezer and all samples were frozen within 24 h of sample collection. Cryopreserved samples were shipped to Toronto, Canada on dry ice, where the cryovials were stored at −80 °C without freeze-thaw until batch analyte quantification.

*Biomarker quantification.* Plasma analyte quantification was performed using reagents from the same lot (R&D Systems) and was completed within three weeks. A detailed protocol for sample quantification has been described by our group previously[41].

The following 11 biomarkers were quantified: sTREM-1, Interleukin-6 (IL-6), Interleukin-8 (IL-8), Soluble Tumour Necrosis Factor Receptor-1 (sTNFR1),

Chitinase-3-like protein 1 (CHI3L1), Interferon Gamma-Induced Protein 10 (IP-10), Soluble fms-like tyrosine kinase 1(sFlt-1), Angiopoietin 1 (Ang-1), Angiopoietin 2 (Ang-2), Soluble Intercellular Adhesion Molecule 1 (sICAM-1) and Soluble Vascular Cell Adhesion Molecule 1 (sVCAM-1). An additional biomarker, Granzyme B, was also quantified, but omitted from statistical analyses, as its values were below the detection limit in 838 of the 2,084 children (40.2%).

*Multiple imputation.* Variables used in multiple imputation model included: hospital death, duration of follow-up, abscondment, age, sex, LODS score, temperature, oxygen saturation, heart rate, lactate, time to evaluation by medical doctor, malaria status, HIV status, pneumonia diagnosis, log-transformed biomarker values (sTREM-1, Ang-2, sFlt-1, sTNFR1).

*Literature review.* We searched PubMed without date restrictions for publications until June 26, 2020 with the search terms "strem-1", "strem 1", "trem-1", "trem 1", "strem1", "trem1", "strem", "trem", OR (soluble AND triggering AND receptor AND myeloid), restricted to humans in the infant and child age group and to publications in English. Among 51 articles, 15 reported on diagnostic or prognostic studies assessing sTREM-1 plasma levels in children at risk of sepsis (Table S1). Our study is the largest prospective study to date to evaluate the ability of sTREM-1 to predict mortality, and the only study performed in unselected febrile children between 2 months and 5 years of age presenting to a referral hospital in a low-resource setting.

### Diagnostic definitions

*P. falciparum malaria.* Malaria diagnosis was based on both the detection of *P. falciparum* histidine rich protein 2 (HRP2) and pan-malaria lactate dehydrogenase (pLDH) by a rapid diagnostic test (RDT) (First Response Malaria Ag. HRP2/pLDH Combo Rapid Diagnostic Test, Premier Medical Corporation Limited, India) and visualization of parasites using light microscopy of Field's stained thick peripheral blood smear[46].

*HIV.* HIV diagnosis was confirmed by two antibody-based RDTs and quantitative polymerase chain reaction (PCR). A sample was considered seronegative if the Alere Determine® RDT (Waltham, MA, USA) was negative. Positive results were confirmed with a second RDT from a different manufacturer (Chembio, Medford, NY, USA). The sample was considered seropositive if both results were positive. Samples were considered indeterminate if the antibody tests were discordant. All seropositive and indeterminate samples were tested by PCR (Abbott Laboratories Ltd.). Children >24 months old were considered HIV infected if they were seropositive, or if they had indeterminate serology and detectable HIV RNA. Infants ≤24 months old were considered HIV infected if they had either seropositive or indeterminate serology and had detectable HIV RNA.

*Pneumonia.* Pneumonia was classified according to the World Health Organization (WHO) Integrated Management of Childhood Illness (IMCI) definition[47]. The definition included a history of cough or difficulty breathing, and if present, respiratory rate >50 breaths/minute if ≤12 months or >40 breaths/minute if >12 months old. Severe pneumonia was defined as presence of pneumonia and any general danger sign.

**Reporting summary.** Further information on research design is available in the Nature Research Reporting Summary linked to this article.

## Data availability
The data supporting the findings from this study are available within the article and its supplementary information. Source data are provided with this paper.

## Code availability
Full SATA code used in the statistical analyses and figure generation in this manuscript are available from the authors on reasonable request.

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

## Acknowledgements

We thank all the children and their caregivers from the Jinja Regional Hospital for participation in the prospective observational study, as well as all healthcare providers and medical students who assisted with study enrolment and patient follow-up. This work was supported by a Collaborative Research Agreement Grant from Intellectual Ventures/Global Good (K.C.K. and W.C.L.), the Canadian Institutes of Health Research (CIHR) Foundation grant FDN-148439 (K.C.K.), the Canada Research Chair Program (P.J. and K.C.K.), a CIHR Banting fellowship (AL), CIHR Postdoctoral Research Fellowship (A.L.C., M.H.), the Bill and Melinda Gates Foundation Trust through Intellectual Ventures/Global Good and donations from Kim Kertland and the Tesari Foundation.

## Author contributions

A.L. conceived and designed the study, performed the experiments, analysed and interpreted data, did the literature search, wrote the first draft of the report and contributed to all revisions. A.L.C., M.H., R.O.O. and S.N. contributed to patients' recruitment, data collection and clinical management, and contributed to all revisions. M.R.G. and K.Z. contributed to performing the experiments and to all revisions. B.d.C. analysed and interpreted data and contributed to all revisions. P.J. designed the study, analysed and interpreted data, contributed to the literature search, wrote the first draft of the report and contributed to all revisions. D.B., W.C.L. and K.C.K. conceived and designed the study, contributed to the experiments, interpreted data and contributed to all revisions. All authors reviewed and approved the final version of the report.

## Competing interests

P.J. serves as unpaid member of steering group or executive committee of trials funded by Abbott Vascular, Astra Zeneca, Biotronik, Biosensors, St. Jude Medical, Terumo and The Medicines Company, has received research grants to the institution from Appili Therapeutics, Astra Zeneca, Biotronik, Biosensors International, Eli Lilly, The Medicines Company, and honoraria to the institution for participation in advisory boards and/or consulting from Amgen, Ava and Fresenius, but has not received personal payments by

any pharmaceutical company or device manufacturer. K.C.K., W.C.L. and A.L.C. are named inventors on a patent "Biomarkers for early determination of a critical or life-threatening response to illness and/or treatment response" held by University Health Network. W.C.L. has an Outcome Predictive Tool in Sepsis (OPTIS) patent pending. The remaining authors declare no competing interests.
