## [Peer Review File · Nature Communications]

REVIEWER COMMENTS

Reviewer #1 (Remarks to the Author):

This is a novel contribution on the use of sTREM-1 at the triage levels for the diagnosis of sepsis in children. I have some concerns for this submission.

- There is lack of clarity in the Methods section on the criteria for the diagnosis of sepsis and of malaria. Both diagnoses are used interchangeably. I do understand that malaria is a common problem in the setting of the authors but one published study will have an impact outside their country. To this end, the specific plasmodia causing sepsis need to be provided. Furthermore, the authors need to change their title for malaria instead of sepsis.
- Why is mortality censored at 7 days and not at a later time point?
- The criteria to define the three colors/strata of risk classification are not clear. The authors need to become precise.
- Sepsis in children is driven through macrophage activation. Is this the case here? Ferritin levels need to be reported.
- What were the criteria for the selection of the 11 biomarkers?

Reviewer #2 (Remarks to the Author):

Thank you for the opportunity to review this interesting manuscript. The authors have explored the prognostic utility of several biomarkers with the goal of assessing their potential to support triage decisions made in community settings in children with fever. There are several strengths of this study including a large sample size and evaluation of several different biomarkers. The need for such a triage tool is well stated, and the goal laudable. The willingness of the authors to share their data and their statistical code also increase transparency and utility of the data. Despite these strengths, there are several methodological details that should be attended to. Most of these can be addressed with alternative ways of examining the data, although there are some aspects of the data collection that insert a small amount of uncertainty into the findings.

i) This study is written and prepared as though it is the derivation and validation of a risk model and the authors have carefully followed guidance in a reference that pertains to risk models in medicine

which would require internal and external validation. However, the study is essentially an assessment of the diagnostic accuracy (prognostic accuracy) of individual biomarkers. As such, the approach to having a derivation and validation cohort seems unnecessary. It would be reasonable for derived cut-points of a diagnostic test to be validated in an external cohort, but that does not seem to be the goal of the study and is certainly not emphasized. Indeed, the cutoffs reported on page 6 were based on pooled data. As such, the split between derivation and validation cohorts seems unnecessary.

ii) At no point do the authors present the ROC curve for the different biomarkers. The shape of the curve and distribution of cases along the curve can be informative, particularly because the slope of the ROC is equivalent to the LR for that particular decision-making threshold. Since this project is based on comparing the AUROC among biomarkers, displaying the curves might be considered essential.

iii) The use of listwise deletion of cases that absconded and for whom vital status is unknown is suboptimal, especially as this is more than about 10% of the cohort. Even in the setting of missing not at random (NMAR), the use of multiple imputation can reduce the bias. Indeed, when there is bias in missingness of outcomes, which is likely in this case, the listwise deletion has much bias that is not considered. The more explanatory variables used in the multiple imputation, the less biased the analysis becomes.

iv) If logistic regression is used for estimating the AUC, then it would be sensible to consider whether the association between biomarker levels and outcomes is 'linear'. Modeling the association as non-linear with, for example, cubic splines might identify clear cut points in the biomarkers.

v) Given the comprehensive approach to modeling these data, is there perhaps a biomarker panel that outperforms an individual marker? There are an increasing number of low cost devices able to measure several biomarker levels simultaneously that might be relevant for this setting.

vi) It is unclear why cases that absconded AFTER 7 days were excluded. Since they survived to the censoring time, there does not seem to be a reason for exclusion. If it is a decision based on local ethical considerations, this should be stated.

vii) The information about the missing screening logs is difficult to handle. On one hand, it makes sense to report this occurred. On the other hand, the local recollection is not likely very accurate. I would suggest not trying to make numeric estimates, and not including this in the CONSORT diagram.

viii) In the supplementary tables, many comparisons are provided, along with p-values. The p-values might be significant because of the large sample size, not because of clinically meaningful differences. It might be better to show the magnitude of the difference with confidence intervals rather than an odds ratio (which is difficult to interpret in this context) and p-values.

ix) The choice of likelihood ratios of 0.1 and 10 for cut points on the biomarkers are reasonable, but it would be very helpful to understand the 'cost' of a false positive and a false negative. This might suggest an emphasis on rule-out, or on rule-in that prompts the use of different levels. This is especially the case because 40% of cases are 'yellow' – or in the grey zone of decision making. It might be possible to set thresholds differently so fewer cases are decision dilemmas.

- x) The choice for subgroup analysis for the survival is ok, but it would be better to formally test for an interaction between biomarker cut points and subgrouping variable to see if there really is a difference between subgroups or not.
- xi) In table 2, the AUC for Ang-1 is less than 0.5. This would suggest that the analysis is 'inverted' and a higher biomarker level is better, not worse.
- xii) It is unclear why Granzyme B was not considered for analysis. Just because the level was below the lower limit of detection does not mean the biomarker has no utility.
- xiii) The inclusion of cases with a history of fever but no current fever is ok, but it is inconsistent with the title of the manuscript and the aim – which is to help decision making in children who are currently febrile.
- xiv) Please provide information on the number of cases for whom blood was not collected on arrival to the emergency department but was instead collected the next morning. It is possible that treatments given overnight might have affected the biomarker levels.
- xv) Please provide a brief discussion of how the included cohort is similar to, or different from, the population in which the test would be used.
- xvi) The authors are commended for providing their sample size estimates, but they do not match the analyses done. This would suggest some differences between the planned approach and what was finally done. Please provide a brief explanation why the two do not match.
- xvii) In the descriptive tables where continuous variables are reported as mean and standard deviation, lactate levels are presented as, I assume, median and IQR. Please label this appropriately.
- xviii) In Table 4, it is surprising that temperature differed between absconded and discharged patients when the effect size was so small (a difference of 0.2°C with a common SD of 1.2°C). Similarly, for other variables the p-values emphasize differences that do not seem to be well supported by the presented descriptive data.
- xix) The external validation calibration plots suggest the biomarker under-predicts mortality. Can the authors explain this? What is the implication for interpreting the results?
- xx) Abstract line four, there appears to be an error. The hypothesis is that the biomarkers identify children at risk of all cause mortality, not all cause sepsis.
- xxi) Overall, it is unclear whether this paper is about sepsis risk, as described in the introduction, or just about mortality risk. There is no cause of death provided to know if sepsis was the precipitating factor. It should be clarified whether sepsis is a truly relevant consideration.

Thank you again for the opportunity to review this manuscript.

Christopher J Lindsell, PhD

Reviewer #3 (Remarks to the Author):

This is a prospective study of febrile children with suspected serious infections evaluated in the Jinja Hospital in Uganda who had plasma samples obtained and assessed against clinical outcomes (mortality). The study was based on two sequential cohorts of children (a derivation cohort followed by a validation cohort). Based on a panel of 11 biomarkers of immune stimulation and endothelial activation, the authors found that the Soluble Triggering Receptor Expressed on Myeloid cells-1 (sTREM-1) had the best ROC and predictive value for inpatient and post-discharge mortality, and recommend that as a rapid screening and triage test.

There are several problems with this approach. Firstly, we are told nothing about the standard of care in this hospital and if the protocols for suspected sepsis or malaria or other serious illnesses in HIV positive subjects for example, were standardized or not? The fundamental premise here that there is a common pathway for sepsis and children at risk of mortality which can be detected early at presentation by a biomarker must be matched against alternative forms of triage and risk characterization. Did the authors undertake any form of standardized clinical risk scoring other than the LODS at admission. It seems that children who died were significantly more hypoxic and hypothermic at admission. Did any of the biomarkers outperform clinical triage and treatment-adjusted outcomes?

It would have helped to see some standard biomarkers such as CRP, AGP in comparative evaluation against the 11 biomarker panel. Could any of the biomarkers have worked better in combination with clinical features at admission, given that the bulk of the differentiation of deaths appears to be early?

Reviewer #4 (Remarks to the Author):

Major comments:

1. The study cohort was divided into a derivation cohort and an external validation cohort based on enrollment date. Such a strategy is crucial for developing and validating a data-driven predictor or classifier, where overfitting can be a serious issue and results can be overly optimistic without

independent validation. For this particular study, I would appreciate some clarifications on what exactly was derived from the derivation cohort and subsequently validated in the validation cohort. It seems that the 11 biomarkers under investigation were pre-defined (right?), and I guess the point of derivation/validation was to identify and validate the best biomarker with the largest AUC, which turned out to be sTREM-1? Was this planned prospectively?

2. In Table 2 and elsewhere, you have presented results of an “internal validation” analysis. What exactly do you mean by “internal validation”? Does this refer to a cross-validation procedure in which a prediction is made for each subject based on independent data from other subjects in a training set? However, the ROC analysis for a given biomarker does not require any training at all. So I don’t see the point of internal validation, and I don’t understand why the results are not identical (I see they are very similar) for derivation and internal validation in Table 2 and Supplementary Tables 7-9. Please provide a rationale for the internal validation or remove it from the paper.

3. The Kaplan-Meier curves in Figure 3 are quite informative as they provide estimates of mortality rates that appropriately account for loss to follow-up. However, I find the associated text (L188-197) less informative for several reasons. First, “8.32 times more likely to die” is not an accurate interpretation of a hazard ratio (HR) of 8.32. Second, any HR is based on a proportional hazards assumption that is almost certainly false. Third, when correctly interpreted, the HR is just not very meaningful to non-statisticians. Why not just provide and compare 7-day mortality rates with confidence intervals in the text? The text can also include p-values for comparing different subgroups. If you really want to show HRs, they can be kept in Figure 3.

4. For children with malaria, you have indicated that “Cox models were unstable” and you therefore used Poisson regression to estimate rate ratios. What was the issue with Cox models? Not enough events? What was the rationale for using Poisson regression? Please provide a justification (e.g., references) for using Poisson regression in this setting, or else remove this analysis.

5. [Optional] Table 1 shows that a number of baseline characteristics (other than biomarkers) are predictive of vital status at 7 days. It is of interest to see how sTREM-1 (and other biomarkers) may add to those baseline characteristics for predicting vital status at 7 days. A simple way to look at this would be comparing predictions based on logistic regression models that include the variables in Table 1 with or without sTREM-1 (and possibly other biomarkers).

Minor comments:

1. L128-131: “An additional 376 children were excluded...as they had absconded or were transferred before or after 7 days (Fig. 1).” I understand that their vital status at 7 days could not be ascertained if they were lost to follow-up before 7 days. I don’t understand what the problem is with those children who were lost to follow-up AFTER 7 days. Don’t we know already that they were alive at 7 days?

2. L183-186: Could you explain somewhere how the calibration was done for Supplementary Figures 2 and 3? This is not trivial because some subjects were lost to follow-up.

3. L326-327: The AUROC can be, and usually is, estimated nonparametrically. Why do you use logistic regression to estimate the AUROC?

4. L359-360: "...internally validated...based on 500 bootstrap samples..." This doesn't make sense to me. Bootstrap is a general approach to variance estimation and inference. I still don't know what the internal validation was.

5. L376-379: A multiple imputation approach was used to deal with missing data on vital status at 7 days, "with all baseline characteristics and in-hospital death up to 7 days as variables in the imputation model to create 20 imputed datasets." I presume that "in-hospital death up to 7 days" was treated as the response variable, right? Could you specify the baseline variables used in the imputation model? Are they the same ones listed in Table 1?

6. Table 1: Are these results based on univariate analyses (logistic regression of vital status on one baseline variable at a time)?

RESPONSES TO REVIEWER COMMENTS

Reviewer #1:

This is a novel contribution on the use of sTREM-1 at the triage levels for the diagnosis of sepsis in children. I have some concerns for this submission.

COMMENT #1: There is lack of clarity in the Methods section on the criteria for the diagnosis of sepsis and of malaria. Both diagnoses are used interchangeably. I do understand that malaria is a common problem in the setting of the authors but one published study will have an impact outside their country. To this end, the specific plasmodia causing sepsis need to be provided. Furthermore, the authors need to change their title for malaria instead of sepsis.

RESPONSE #1: Thank you for the clarification requests. All malaria infections in this study were caused by *Plasmodium falciparum*. This diagnosis was based on confirmation by an expert microscopy as well as by Rapid Diagnostic Tests (please refer to the detailed description in the supplemental methods). As requested, we changed the text and figure legends from “malaria” to “*P. falciparum* malaria”.

To qualify for enrollment, children had to have a history of fever and require hospital admission for a suspected infection. Please note that the major difference in the definition of SIRS (systemic inflammatory response syndrome, used in the definition of sepsis) between adults and children is that the diagnosis of pediatric SIRS requires that temperature or leukocyte abnormalities be present (i.e., SIRS should *not* be diagnosed if a pediatric patient exhibits only elevated heart and respiratory rates)¹. As such, all children met the sepsis definition.

In addition, the majority of children met at least one additional pediatric SIRS criteria (please see table below). Therefore, our cohort represents a pediatric sepsis cohort. Sepsis is a dysfunctional host immune response to infection (any infection) that contributes to multi-organ dysfunction and death. This corresponds to our focus on the detection of host immune and endothelial activation early in the course of illness to identify children at risk of infection-related death. Of note that malaria is an infectious etiology causing sepsis and is among the leading causes of sepsis worldwide². For these reasons, we have not changed the main title of our manuscript from sepsis to malaria. However, as suggested and to avoid any confusion we clarified that *P. falciparum* malaria is classified as a sepsis syndrome according to current definition of sepsis.²

Vital sign abnormality*	Proportion of children
History of fever	100%
Abnormal heart rate	59.2%
Abnormal respiratory rate	94.8%
Abnormal systolic blood pressure	23.1%
≥2 pediatric SIRS	94.8%
≥3 pediatric SIRS	66.1%
4 pediatric SIRS	11.5%

*Abnormal vital signs are age-dependent and are defined as follows¹:

1. Abnormal heart rate: >180/<90 beats/min (if age 1 month-1 year) or >140 beats/min (if age 2-5 years)
2. Abnormal respiratory rate: >34 breaths/min (if age 1 month-1 year) or >22 breaths/min (if age 2-5 years)
3. Abnormal systolic blood pressure: <100 mmHg (if age 1 month-1 year) or <94 mmHg (if age 2-5 years)

*Leukocyte abnormalities could not be ascertained due to a lack of hematology laboratory services (not within routine care practices in health care facilities in Uganda).

COMMENT #2: Why is mortality censored at 7 days and not at a later time point?

RESPONSE #2: We censored mortality at 7 days and not at a later time point because 7-day mortality provided the most clinically meaningful, plausible, and feasible time frame for ascertaining the primary outcome. Most deaths in our cohort (96%) occurred within the first 7 days (please see table below). The timing of mortality in our cohort is similar to other large pediatric cohorts in Africa. For example, the majority of pediatric deaths due to infectious etiologies also occur early after presentation (i.e.: malaria, AQUAMAT³, where 66.5% of death occurred within 24 hours).

Day of death	# of deaths	Total deaths	Cumulative %
0	41	41	41.4
1	44	85	85.9
2	6	91	91.9
3	1	92	92.9
4	2	94	94.9
6	1	95	96.0
11	1	96	97.0
12	1	97	98.0
13	1	98	99.0
17	1	99	100

Deaths significantly after 7 days (i.e.: a later time point) would have likely been due to causes unrelated to initial hospital presentation (i.e.: other than presenting infectious etiology).

COMMENT #3: The criteria to define the three colors/strata of risk classification are not clear. The authors need to become precise.

RESPONSE #3: Thank you for this request. Please note that in the Methods subsection “Statistical analysis” in paragraph 1, page 14, we have provided the following detailed explanation:

“we identified the sTREM-1 levels in pg/mL that were associated with a negative likelihood ratio (LR-) of 0.10 and a positive likelihood ratio (LR+) of 10. A LR- of 0.10 indicates that it was 10 times less likely to find a sTREM-1 concentration at clinical presentation, lower than the associated cutoff in children who subsequently died as compared with those who survived. A LR+ of 10 indicates that it was 10 times more likely to find a sTREM-1 concentration equal to or higher than the associated cutoff in children

who subsequently died as compared with those who survived.⁴ The pre-specified targets of 0.10 for the LR- and 10 for the LR+ are considered to be associated with large, often conclusive changes from pre-test to post-test probabilities.⁵ Likelihood ratios were preferred over predicted risks to derive cutoffs for the projected rapid triage test, as they did not depend on variations in the underlying risk of death in the studied population. Children with sTREM-1 values below the cutoff associated with a LR- of 0.10 were classified as low risk (green zone), children with sTREM-1 values above the cutoff associated with a LR+ of 10 as high risk (red zone). Remaining children were classified as intermediate risk (yellow zone). For descriptive purposes, we also estimated LR- and LR+ for different cutoffs of sTREM-1 (see Supplementary Information).”

In addition, Figure 2 legend provides the following simplified explanation:

“Negative and positive Likelihood Ratios (LRs) in the derivation and validation cohorts combined were used to risk-stratify febrile children: “green” zone: low risk (LR- of 0.10, sTREM-1 <239 pg/mL), “yellow” zone: refer and monitor (sTREM-1 ≥239 pg/mL and <629 pg/mL), “red” zone: urgent admission/support (LR+ of 10, sTREM-1 ≥629 pg/mL)”

As requested, for clarity, we have now added a reference to our detailed methods section explaining the color risk stratification in Results subsection 3, entitled “*Risk stratification based on sTREM-1 level*” page 6, paragraph 2 of this subsection.

COMMENT #4: Sepsis in children is driven through macrophage activation. Is this the case here? Ferritin levels need to be reported.

RESPONSE #4: Thank you for the insightful comment. Indeed, sepsis in both children and adults is in part driven by macrophage activation. Macrophage activation is among one of many dysregulated responses during acute infection-related critical illness. On this note, our lead biomarker, sTREM-1, is a marker of myeloid cell activation (Triggering Receptor Expressed on Myeloid cells). As such, sTREM-1 plasma levels also represent activation of monocytes and macrophages.

Collectively, our focus was on pathways related to immune and endothelial activation implicated in the pathogenesis of infection-related multi-organ failure and death in many prospective studies⁶⁻¹⁵. Markers of these pathways play mechanistic roles in organ dysfunction and death related to sepsis (e.g. Angpt-Tie2 axis, VEGF pathway (e.g. sFlt-1), TREM-1) and are also druggable, meaning that interventions that target these pathways may improve outcome.¹⁶

Ferritin is an acute phase protein and, similar to CRP and PCT, is non-specifically increased during infection. In settings where anemia is prevalent, such as in resource limited settings similar to Jinja, Uganda, defining hyperferritinemia is not well studied¹⁷. Also, we are not aware of data in children indicating that measuring ferritin would provide insight into the pathways of immune and endothelial activation, the focus of our study.

However, please note that we did compare two commonly used acute phase proteins (i.e.: C-reactive protein [CRP] and Procalcitonin [PCT]). In a subset of patients with available data for CRP and PCT, the predictive performance was poor relative to the lead marker of immune activation (please see Response #3 to Reviewer #3).

COMMENT #5: What were the criteria for the selection of the 11 biomarkers?

RESPONSE #5: Thank you for the request for this clarification. As mentioned in the reply immediately above, the focus of our study was the characterization of immune and endothelial activation in severe infection. These pathways have been implicated in the pathogenesis of infection-related multi-organ failure and death in many prior prospective studies⁶⁻¹⁵. Please note that these same references are included in the Introduction section of the manuscript (page 3, paragraph 2).

To this end, our laboratory has validated these markers and have extensive data to support the reliable measurement of these circulating proteins^{8,18,19}. To be considered for inclusion in our multiplex panel, the markers had to have a biologic link to severe infection (i.e.: immune or endothelial activation) and be able to be combined in a commercially available custom-built assay format that was reproducible, reliable, and potentially scalable. We also tested the markers on a near-patient-care platform (EllaTM)¹⁸ to provide data on the use of a platform that was more scalable and automated to facilitate real-time triage in future validation studies.

Of note, although many of the markers we included in our study can predict mortality in adults in high resource settings^{6,10-12,14}, this study represents the first large-scale “head to head” comparison in a febrile pediatric population in a low resource setting, where the burden of severe infections is the highest².

Reviewer #2:

Thank you for the opportunity to review this interesting manuscript. The authors have explored the prognostic utility of several biomarkers with the goal of assessing their potential to support triage decisions made in community settings in children with fever. There are several strengths of this study including a large sample size and evaluation of several different biomarkers. The need for such a triage tool is well stated, and the goal laudable. The willingness of the authors to share their data and their statistical code also increase transparency and utility of the data. Despite these strengths, there are several methodological details that should be attended to. Most of these can be addressed with alternative ways of examining the data, although there are some aspects of the data collection that insert a small amount of uncertainty into the findings.

Thank you very much for the complimentary comments. Please see our replies to your concerns below.

COMMENT #1: This study is written and prepared as though it is the derivation and validation of a risk model and the authors have carefully followed guidance in a reference that pertains to risk models in medicine which would require internal and external validation. However, the study is essentially an assessment of the diagnostic accuracy (prognostic accuracy) of individual biomarkers. As such, the approach to having a derivation and validation cohort seems unnecessary. It would be reasonable for derived cut-points of a diagnostic test to be validated in an external cohort, but that does not seem to be the goal of the study and is certainly not emphasized. Indeed, the cutoffs reported on page 6 were based on pooled data. As such, the split between derivation and validation cohorts seems unnecessary.

RESPONSE #1: As pointed out in the methods section, we updated the model based on the pooled data of all children with available plasma of derivation and validation cohorts combined only after we had established that internal and external validations were successful to make full use of all available information. As shown in Supplementary Table 9 and indicated in the results section (page \$, paragraph \$), likelihood ratios were similar between derivation, internal and external validation for all cutoffs.

We strongly feel that the prespecified internal and external validation add considerably to the credibility of our analysis, as sTREM-1 is top ranked in all 3 analyses, and areas under the receiver operating curves of individual biomarkers are very similar between derivation, internal and external validation (Table 1). In addition, estimated likelihood ratios are comparable between derivation, internal and external validation (Supplementary Table 9).

As indicated in the introduction of Reviewer #4, the chosen approach prevents overfitting, and also establishes temporal validity of the chosen biomarker and its cutoffs.

COMMENT #2: At no point do the authors present the ROC curve for the different biomarkers. The shape of the curve and distribution of cases along the curve can be informative, particularly because the slope of the ROC is equivalent to the LR for that particular decision-making threshold. Since this project is based on comparing the AUROC among biomarkers, displaying the curves might be considered essential.

RESPONSE #2: Thank you for this suggestion. As requested, below please find combined AUROC curves for all biomarkers in the (A) derivation (n=1176), and (B) validation (n=908) sets of the complete cohort for in children included in biomarker comparative performance

(data correspond to AUROC and 95% CI presented in main manuscript Table 2). These figures are also included in the revised Supplementary Information (**Supplementary Fig.1a-b**).

(A)

(B)

COMMENT #3: The use of listwise deletion of cases that absconded and for whom vital status is unknown is suboptimal, especially as this is more than about 10% of the cohort. Even in the setting of missing not at random (NMAR), the use of multiple imputation can reduce the bias. Indeed, when there is bias in missingness of outcomes, which is likely in this case, the listwise deletion has much bias that is not considered. The more explanatory variables used in the multiple imputation, the less biased the analysis becomes.

RESPONSE #3: We agree that multiple imputation can reduce bias and have used this as main strategy in our study (please refer to the Methods subsection “*Statistical analysis*”). In this section, we describe how multiple imputation was performed:

“To account for missing vital status in children who were transferred or absconded before 7 days we used multiple imputation,²⁰ with all baseline characteristics and in-hospital death up to 7 days as variables in the imputation model to create 20 imputed datasets. When deriving time-to-event curves for descriptive purposes, however, we censored children who were transferred to another hospital or absconded before 7 days at the day of transfer or abscondment”

Variables used in multiple imputation included those that were associated with mortality and included: the Lambaréné Organ Dysfunction Score (LODS, a simple, validated severity of illness score²¹), lactate, hemoglobin, time to MD (defined as time from ER presentation to time of evaluation by physician, with children who clinically appeared sick based on clinical judgement having been seen earlier; a shorter time to MD evaluation correlated with LODS/death, *P. falciparum* malaria diagnosis (children diagnosed with *P. falciparum* malaria had a better outcome as they received treatment targeted against the appropriate pathogen), World Health Organization (WHO) Integrated Management of Childhood Illness (IMCI) diagnosis of pneumonia²² (a calculated variable based on IMCI criteria, defined by the presence of a cough and a respiratory rate above cut off for age >/< 12mths), age (in months), and sex (variable “male”).

Of note, children who absconded, were transferred or for whom vital status was unknown were only excluded from the analysis of the comparative performance of 11 biomarkers, as biomarkers originally were not measured in this subgroup, following a decision of the investigators made in 2016.

sTREM-1 and sFlt-1, which ranked first and second in the analysis of the derivation cohort, sTNFR1, which ranked second after sTREM-1 in the previously reported study in febrile adults in Tanzania, and Ang-2, which was the biomarker used for sample size considerations of the current study, were subsequently quantified using identical methods in the remaining 376 children with available plasma who were transferred or absconded.

The main analysis on the prognostic performance of sTREM-1 is therefore based on all 2460 children with available plasma. This represents 98.3% of the included cohort.

COMMENT #4: If logistic regression is used for estimating the AUC, then it would be sensible to consider whether the association between biomarker levels and outcomes is ‘linear’. Modeling the association as non-linear with, for example, cubic splines might identify clear cut points in the biomarkers.

RESPONSE #4: We used a nonparametric receiver operating characteristic analysis to estimate the area under the curve. We now clarify this in the Methods Statistical analysis section, on page 14, paragraph 2:

“For the analysis of the comparative performance of all 11 biomarkers, we used a non-parametric approach to determine the AUROC as a measure of discrimination between children who died from any cause up to 7 days and children who survived, and ranked biomarkers according to the estimated AUROC in children of the derivation cohort with an available plasma sample who had died or were regularly discharged from hospital (n=1176), but were neither transferred to another hospital nor absconded.²³”

COMMENT #5: Given the comprehensive approach to modeling these data, is there perhaps a biomarker panel that outperforms an individual marker? There are an increasing number of low cost devices able to measure several biomarker levels simultaneously that might be relevant for this setting.

RESPONSE #5: Our goal was to devise the simplest triage tool possible to be used in community settings by minimally trained staff, without compromising feasibility of risk stratification. As such, our goal was to devise ideally a univariate model that would be simple to implement as a semi-quantitative rapid triage test with a 3-level outcome (low-risk green zone, intermediate risk yellow zone, high risk red zone).

To address this reviewer’s comment, we analysed biomarkers individually and in combinations of the top-performing biomarkers, sTREM-1 as a single biomarker offered the best discrimination that was statistically superior to a 2 and 3 variable model (derivation cohort, please see data analysis below). The simple, parsimonious model of using only sTREM-1 remains the preferred model.

Below, please find results of stepwise addition of one biomarker at a time to assess whether the addition of next best performing biomarkers would improve the prediction of 7-day mortality. Biomarkers were added in order of best improvement in AUROC relative to a biomarker model with one less biomarker included). Data are presented for (A) the derivation cohort and (B) validation cohort.

(A) Derivation cohort:

Model	Derivation (n=1176) AUROC (95% CI)	P-value
1 biomarker (sTREM-1)	0.893 (0.843-0.944)	-
2 biomarkers (+IL-6)	0.905 (0.854-0.957)	0.397
3 biomarkers (+Ang2)	0.920 (0.874-0.966)	0.077
4 biomarkers (+IP10)	0.929 (0.887-0.972)	0.052
5 biomarkers (+IL-8)	0.931 (0.888-0.975)	0.579
6 biomarkers (+Ang1)	0.936 (0.897-0.975)	0.273
7 biomarkers (+sTNFR1)	0.940 (0.904-0.976)	0.240
8 biomarkers (+sICAM1)	0.940 (0.904-0.976)	0.480
9 biomarkers (+sVCAM1)	0.940 (0.904-0.976)	0.760
10 biomarkers (+CHI3L1)	0.940 (0.903-0.976)	0.844
11 biomarkers (+sFlt-1)	0.939 (0.900-0.978)	0.870

(B) Validation cohort:

Model	Validation (n=908) AUROC (95% CI)	P-value
1 biomarker (sTREM-1)	0.901 (0.856-0.947)	-
2 biomarkers (+IP10)	0.917 (0.879-0.956)	0.026
3 biomarkers (+Ang1)	0.922 (0.882-0.962)	0.441
4 biomarkers (+IL-6)	0.929 (0.889-0.968)	0.442
5 biomarkers (+sVCAM)	0.930 (0.891-0.969)	0.208
6 biomarkers (+IL-8)	0.929 (0.887-0.971)	0.772
7 biomarkers (+Ang2)	0.929 (0.888-0.971)	0.941
8 biomarkers (+CHI3L1)	0.929 (0.888-0.971)	0.748
9 biomarkers (+sICAM1)	0.929 (0.888-0.971)	0.686
10 biomarkers (+sFlt-1)	0.929 (0.887-0.971)	0.259
11 biomarkers (+sTNFR1)	0.929 (0.886-0.971)	0.414

*Note: The 11 biomarker model includes all biomarkers (sTREM-1, IL-8, Ang2, CHI3L1, sTNFR1, IL-6, sICAM-1, sVICAM-1, IP10, Ang1)

These data are in agreement with our findings in febrile adults in Tanzania where, as in this study, no combination outperformed sTREM-1 alone; and sTREM-1 significantly improved the predictive performance of clinical scores¹⁹.

COMMENT #6: It is unclear why cases that absconded AFTER 7 days were excluded. Since they survived to the censoring time, there does not seem to be a reason for exclusion. If it is a decision based on local ethical considerations, this should be stated.

RESPONSE #6: As indicated in our response to comment #3 of this Reviewer, children who absconded, were transferred or for whom vital status was unknown were only excluded from the analysis of the comparative performance of 11 biomarkers, as biomarkers originally were not measured in this subgroup, following a decision of the investigators made in 2016. This decision was made at the time even though the vital status up to 7 days was known in those children who absconded or were transferred later than 7 days.

However, sTREM-1 and sFlt-1, which ranked first and second in the analysis of the derivation cohort, sTNFR1, which ranked second after sTREM-1 in the previously reported study in febrile adults in Tanzania, and Ang-2, which was the biomarker used for sample size considerations of the current study, were subsequently quantified using identical methods in the remaining 376 children with available plasma who were transferred or absconded.

The main analysis on the prognostic performance of sTREM-1 is therefore based on all 2460 children with available plasma. This represents 98.3% of the included cohort.

Thank you for this thoughtful comment. Cases that absconded after 7 were excluded as follow-up data were not available for those patients. Due to funding constraints, patients were followed-up only during the course of hospital stay, and not in the community. Additionally, we did not have local IRB approval nor the financial or logistic means for home visits after abscondment or discharge. Please also see our response to Reviewer #1 comment #2 (“*Why is mortality censored at 7 days and not at a later time point?*”).

COMMENT #7: The information about the missing screening logs is difficult to handle. On one hand, it makes sense to report this occurred. On the other hand, the local recollection is not likely very accurate. I would suggest not trying to make numeric estimates, and not including this in the CONSORT diagram.

RESPONSE #7: We thank the reviewer for this suggestion. We removed information about the missing screening log from the CONSORT diagram and legend of Figure 1.

COMMENT #8: In the supplementary tables, many comparisons are provided, along with p-values. The p-values might be significant because of the large sample size, not because of clinically meaningful differences. It might be better to show the magnitude of the difference with confidence intervals rather than an odds ratio (which is difficult to interpret in this context) and p-values.

RESPONSE #8: As suggested, we removed all p-values from the supplementary tables.

To address the reviewer’s comment, we calculated the difference in proportions with confidence intervals using generalized linear models with binomial family and identity link. However, the model did not converge for several of the baseline variables analysed. We thus present odds ratios to express the magnitude of the difference in baseline variables, as in the original submission.

COMMENT #9: The choice of likelihood ratios of 0.1 and 10 for cut points on the biomarkers are reasonable, but it would be very helpful to understand the ‘cost’ of a false positive and a false negative. This might suggest an emphasis on rule-out, or on rule-in that prompts the use of different levels. This is especially the case because 40% of cases are ‘yellow’ – or in the grey zone of decision making. It might be possible to set thresholds differently so fewer cases are decision dilemmas.

RESPONSE #9: The rationale for the cutoffs using a negative likelihood ratio (LR) of 0.1 and LR+ of 10 was to clearly distinguish between cases in the “green” zone (no referral) and “red” (definitive urgent referral). The “yellow” zone does not represent a “grey zone”, but rather identifies children who also require triage and evaluation by a trained healthcare professional at a better equipped healthcare facility. Our proposed strategy decreases the referral/admission of children who are at a very low risk of death (green zone) by approximately 50%. In addition, it identifies approximately 10% of children who are in urgent need of immediate care (red zone).

As requested, we now include in the Supplementary Information Table 10 which includes false positive (FP) and false negative (FN) values in the derivation, internal validation, and external validation cohorts for the same sTREM-1 intervals included in Table 9.

COMMENT #10: The choice for subgroup analysis for the survival is ok, but it would be better to formally test for an interaction between biomarker cut points and subgrouping variable to see if there really is a difference between subgroups or not.

RESPONSE #10: The table below provides the only subgroup analyses we performed, by malaria status, as requested by this reviewer. The lower cutoff of 239 pg/mL was selected to ensure an adequate power to rule out children at risk of death, the higher cutoff of 629 pg/mL was selected to ensure adequate power to rule in children at risk of death. The corresponding negative and positive likelihood ratios relevant for the subgroup analyses are highlighted in bold. For the lower cutoff, the test for interaction between negative likelihood ratio and subgroup was 0.37, for the upper cutoff, the test for interaction between positive likelihood ratio and subgroup was 0.16. These negative tests for interactions indicate that the power to rule out children at risk of death of the lower cutoff and the power to rule in children at risk of death of the higher cutoff are consistent between malaria positive and negative children, which in turn underscores the clinical usefulness of biomarker and cutoffs.

		P-value for interaction
Lower cutoff: 239 pg/mL		
Positive likelihood ratio		<0.0001
Malaria +	1.89 (1.78 to 2.01)	
Malaria -	2.58 (2.30 to 2.88)	
Negative likelihood ratio		0.37
Malaria +	0.05 (0.01 to 0.35)	
Malaria -	0.13 (0.06 to 0.31)	
Higher cutoff: 629 pg/mL		
Positive likelihood ratio		0.16
Malaria +	9.51 (7.31 to 12.38)	
Malaria -	13.26 (9.06 to 19.42)	
Negative likelihood ratio		0.042
Malaria +	0.25 (0.14 to 0.46)	
Malaria -	0.50 (0.39 to 0.65)	

In the unlikely case that this reviewer’s comment refers to differences in prognosis between green, yellow and red zones rather than to the subgroup analysis on malaria status, we refer to the omnibus p-values for differences between zones, which were all statistically significant, indicating that green, yellow and red zones are indeed distinct (see our Response #3 to Reviewer #4).

COMMENT #11: In table 2, the AUC for Ang-1 is less than 0.5. This would suggest that the analysis is ‘inverted’ and a higher biomarker level is better, not worse.

RESPONSE #11: Thank you for this correct observation. This is because Angiotensin-1 is constitutively expressed during homeostasis and is a marker of endothelial quiescence. It is as such protective and counters the endothelial activation and microvascular leak induced by Angiotensin-2²⁴. The ratio of angiotensin-2/angiotensin-1 is sometimes used to assess the net impact of dysregulation of this pathway on endothelial function.

COMMENT #12: It is unclear why Granzyme B was not considered for analysis. Just because the level was below the lower limit of detection does not mean the biomarker has not utility.

RESPONSE #12: Thank you for this comment. While most Granzyme B values fell below the limit of detection, in theory, this biomarker may still be important for risk-stratification²⁵⁻³⁰. However, in the context of our study, values below the limit of detection were not interpretable as they could not be expressed on a linear scale and used in logistic regression analysis. Our goal was to identify easily measurable circulating proteins with a wide dynamic range to facilitate reliable triage. Future tests or alternative means of quantifying Granzyme B may offer insight into pathogenesis and prognostic enrichment. However, for our current study, the low abundance of this protein prevented meaningful analysis and outcome prediction.

COMMENT #13: The inclusion of cases with a history of fever but no current fever is ok, but it is inconsistent with the title of the manuscript and the aim – which is to help decision making in children who are currently febrile.

RESPONSE #13: Thank you for this observation. However, we respectfully disagree with the reviewer. Fever syndromes in children are typically intermittent for many documented acute infections. For example, 25-33% of children with documented acute malaria are afebrile when first assessed. Since we are attempting to provide guidance for “fever syndromes” – it would be incorrect to ignore children with a history of fever within 48 hours prior to presentation who continue to be ill enough for their caregiver to bring them for a formal health care evaluation.

Of note, some children with a suspected infection can also present with intermittent *hypothermia*, as per Pediatric CCM guidelines for sepsis diagnosis which includes hypothermia in case definition of sepsis¹.

COMMENT #14: Please provide information on the number of cases for whom blood was not collected on arrival to the emergency department but was instead collected the next morning. It is possible that treatments given overnight might have affected the biomarker levels.

RESPONSE #14: Thank you for this comment regarding timing of blood sample collection. In our healthcare setting this was not a significant issue because all children had blood samples collected at the time of clinical presentation/enrollment and evaluation by a physician. The variable “Time to MD” represents the timing of sample collection, with significant delays observed mainly in patients who presented after 8pm and were not evaluated until the following morning. Of 2,502 enrolled children, 11 presented after working hours (after 8pm), 8 of whom

had follow-up to hospital discharge and were included in the analysis. The median time from presentation to sample collection in children who were included in analysis was 2.6 hours (interquartile range (IQR) 1.0, 4.1). Please refer to Supplementary Information, Patient enrollment section.

In addition, we and others, have previously reported that changes in markers of immune and endothelial activation included in our study (i.e.: Ang-2), if measured daily, can correspond to response to therapy. Although we acknowledge that all plasma proteins have unique circulating half-lives, some markers (i.e.: Ang-2) are high initially and remain high for 18-24 hours, followed by a decline if responding to therapy or improving clinically³¹. Patients whose levels remain at a sustained high level represent a population of high-risk patients and a delayed blood draw in this population would be less likely to alter their high biomarker levels.

COMMENT #15: Please provide a brief discussion of how the included cohort is similar to, or different from, the population in which the test would be used.

RESPONSE #15: The study was designed to be pragmatic and representative of the population in which the intended test would be used (i.e.: triage of fever syndromes in children presenting at formal health care settings such as outpatient clinics and emergency departments). The next steps will require a rapid point-of-care (POC) or “near patient” test for sTREM-1 quantification and a prospective study to assess the utility of a POC quantification of biomarkers for patient risk-stratification would enable early triage (i.e.: decision to admit vs. discharge), appropriate referral (i.e.: consultation with an intensive care unit), and initiation of therapy (i.e.: decision to administer antimicrobials). The other intended population would be the triage of pediatric fever syndromes in community settings, where access to trained healthcare professionals is absent. A 1-hour hand free “near-patient” version is currently available and a POC rapid version is in development.

COMMENT #16: The authors are commended for providing their sample size estimates, but they do not match the analyses done. This would suggest some differences between the planned approach and what was finally done. Please provide a brief explanation why the two do not match.

RESPONSE #16: We included the following explanation in the Methods Statistical analysis section of the main manuscript (page 14, paragraph 1), which is in line with the supplementary appendix:

“The sample size consideration is described in the Supplementary Information. It assumed a nested case-control design, defining in-hospital deaths as cases and using 3 survivors per case as matching controls. The current analysis reflects the original prospective cohort design rather than a nested case-control design.”

COMMENT #17: In the descriptive tables were continuous variables are reported as mean and standard deviation, lactate levels are presented as, I assume, median and IQR. Please label this appropriately.

RESPONSE #17: Continuous variables in descriptive tables in the supplementary material section for lactate are presented as geometric means with 95% reference range. This has been clarified in supplementary table legends.

COMMENT #18: In Table 4, it is surprising that temperature differed between absconded and discharged patients when the effect size was so small (a difference of 0.2°C with a common SD of 1.2°C). Similarly, for other variables the p-values emphasize differences that do not seem to be well supported by the presented descriptive data.

RESPONSE #18: This is indeed due to the large sample size. Again, we followed this reviewer's recommendation and report estimates without p-values throughout the supplementary tables. We also attempted to derive risk differences using generalized linear models with binomial family and identity link. However, the model did not converge for several of the baseline variables analysed. We thus present odds ratios to express the magnitude of the difference in baseline variables, as in the original submission.

COMMENT #19: The external validation calibration plots suggest the biomarker under-predicts mortality. Can the authors explain this? What is the implication for interpreting the results?

RESPONSE #19: As pointed out in the main manuscript, the 7-day mortality was higher in the validation cohort as compared with the derivation cohort.

We expanded the statement addressing this point in the limitations section of the discussion as follows:

“Third, calibration was only modest in the external validation, as the predicted mortality for children in the red zone was lower than observed in this cohort, which likely reflects the trend towards higher mortality in the validation cohort compared to the derivation cohort. However, this does not alter the suggested strategy for triage: children in the red zone would be at high risk of death, regardless of the actual risk of 1 in 4 in the derivation cohort, or 1 in 3 in the validation cohort.”

COMMENT #20: Abstract line four, there appears to be an error. The hypothesis is that the biomarkers identify children at risk of all cause mortality, not all cause sepsis.

RESPONSE #20: We thank the reviewer for noting this error. We revised the abstract, and the sentence now states: *“Here we test the hypothesis that measuring circulating markers of immune and endothelial activation may identify children with sepsis at risk of all-cause mortality”*.

COMMENT #21: Overall, it is unclear whether this paper is about sepsis risk, as described in the introduction, or just about mortality risk. There is no cause of death provided to know if sepsis was the precipitating factor. It should be clarified whether sepsis is a truly relevant consideration.

RESPONSE #21: Thank you for these comments, we have clarified the abstract to reflect that the manuscript focuses on identifying children with sepsis, or impending sepsis, who are at high risk of death, in order to facilitate triage of children with febrile syndromes. The children in our cohort all presented with an acute febrile illness. Infection resulting in sepsis (i.e.: infection-related organ dysfunction) is the most common cause of death in febrile children in sub-Saharan Africa². Approximately 50% of these children had confirmed severe *P. falciparum* malaria, which is one of the causes of sepsis².

Therefore, this study represents sepsis-related mortality. The proximal primary outcome (7-day mortality) also enabled attributing death to the reason for admission (life-threatening infection/sepsis). Please note that other strategies to determine cause of death (i.e.: autopsy) are challenging to perform for cultural and financial reasons. Furthermore, the contributing microbial cause of sepsis (etiology of infection) is not routinely available in resource limited settings due to lack of microbiology laboratories in most referral hospitals.

Thank you again for the opportunity to review this manuscript.
Christopher J Lindsell, PhD

Reviewer #3:

This is a prospective study of febrile children with suspected serious infections evaluated in the Jinja Hospital in Uganda who had plasma samples obtained and assessed against clinical outcomes (mortality). The study was based on two sequential cohorts of children (a derivation cohort followed by a validation cohort). Based on a potential 11 biomarkers of immune stimulation and endothelial activation, the authors found that the Soluble Triggering Receptor Expressed on Myeloid cells-1 (sTREM-1) had the best ROC and predictive value for inpatient and post-discharge mortality and recommend that as a rapid screening and triage test.

COMMENT #1: There are several problems with this approach. Firstly, we are told nothing about the standard of care in this hospital and if the protocols for suspected sepsis or malaria or other serious illnesses in HIV positive subjects for example, were standardized or not?

RESPONSE #1: Thank you for noting this consideration. We agree with the importance to provide detailed information about patient management. Please note that this is summarized in detail in the supplementary information (Table 2: “*Treatments reported at time of presentation*”). Furthermore, the supplementary methods section also provides information about patient enrolment. Briefly, all children were managed by standardized Ugandan National Guidelines for malaria, pneumonia, sepsis, suspected viral hemorrhagic fever.

COMMENT #2: The fundamental premise here that there is a common pathway for sepsis and children at risk of mortality which can be detected early at presentation by a biomarker must be matched against alternative forms of triage and risk characterization. Did the authors undertake any form of standardized clinical risk scoring other than the LODS at admission. It seems that children who died were significantly more hypoxic and hypothermic at admission. Did any of the biomarkers outperform clinical triage and treatment-adjusted outcomes?

RESPONSE #2: This pragmatic prospective study did not intervene with routine patient triage or care. The observational nature of the prospective study enabled studying current local practices in a resource limited setting. The variables collected at the time of hospital presentation were collected for research purposes and not used in routine triage or patient care.

Our goal was to devise the simplest triage tool possible to be used in community settings by minimally trained staff, without compromising feasibility of risk stratification. As such, our goal was to devise ideally a univariate model that would be simple to implement as a semi-quantitative rapid triage test with a 3-level outcome, e.g. from + to +++.

However, as shown in various tables, we collected clinical variables at the time of hospital presentation that would enable calculating 3 different standardized clinical risk scores (LODS³², PEDIA³³, SICK³⁴). We have previously reported that the LODS score performed best in predicting mortality²¹. In addition, the score does not require laboratory studies (i.e.: complete blood count), and as such is the most appropriate potential severity of illness score for low resource settings.

We assessed the predictive performance of sTREM-1 relative to the best clinical score (LODS) and found that its performance was statistically similar (Derivation cohort AUROC for sTREM 0.894 (95% CI 0.843-0.944) versus for LODS 0.907 (95% CI 0.869-0.944), $P = 0.661$;

Validation cohort AUROC for sTREM 0.901 (95% CI 0.856-0.946) versus for LODS 0.912, 95% (CI 0.875-0.949; $P = 0.628$). As such, because sTREM-1 is an unbiased biologic marker that can be quantified at the point of care without the requirement of a skilled healthcare provider clinical assessment, it has the potential to triage children in resource limited settings lacking a trained health care professional in a community setting.

When LODS and sTREM-1 were included in the same logistic regression model, the predictive performance was significantly improved relative to LODS alone in both the Derivation (A) and Validation (B) cohorts. In the derivation cohort (A), the AUROC for LODS alone was 0.907 (95% CI 0.869-0.944) and for LODS plus sTREM-1 was 0.946 (95% CI 0.919-0.973), $P = 0.0003$. In the validation cohort (B), the AUROC for LODS alone was 0.912 (95% CI 0.875-0.949) and for LODS plus sTREM-1 was 0.942 (95% CI 0.903-0.981), $P = 0.006$.

Since this was beyond the scope of the current manuscript, we refrained from including this information in the manuscript, but would of course be ready to do so if requested by the Editors.

(A) Derivation cohort AUROC for LODS and LODS plus sTREM-1

(B) Validation cohort AUROC for LODS and LODS plus sTREM-1

COMMENT #3: It would have helped to see some standard biomarkers such as CRP, AGP in comparative evaluation against the 11 biomarker panel.

RESPONSE #3: Thank you for this anticipated concern. We included C-reactive protein (CRP) and Procalcitonin (PCT) quantification in a subset of 405 children. These children were randomly selected at a 1:4 ratio of children who died versus those who survived the hospital admission (a case-control cohort). A case:control approach with random selection of cases and controls matched at a 1:4 ratio was selected to represent the whole cohort and was chosen to preserve sample and taking funding constraints into consideration. Below please find a breakdown of the children included in the analyses:

	Whole case-control cohort	Included in derivation cohort	Included in validation cohort
Survived	310	183	127
Died	95	43	52
Total	405	226	179

The predicative accuracy of 7-day mortality measured by AUROC of sTREM-1 (the top performing biomarker) relative to CRP and PCT in the whole case:control cohort as well as in the derivation and validation subsets are summarized below:

	AUROC (95% CI) Whole case-control cohort (n=405)	AUROC (95% CI) Derivation cohort (n=226)	AUROC (95% CI) Validation cohort (n=179)
sTREM-1	0.874 (0.834-0.915)	0.872 (0.812-0.932)	0.879 (0.823-0.936)
CRP	0.626 (0.559-0.694)	0.645 (0.547-0.742)	0.597 (0.501-0.693)
PCT	0.688 (0.622-0.754)	0.667 (0.565-0.769)	0.695 (0.605-0.785)

Relative to sTREM-1, CRP and PCT were inferior in all three analyses (whole case control, derivation cohort, and validation cohort, $P < 0.0001$). These data are presented as AUROC graphs below for the (A) Derivation, and (B) Validation cohort.

(A) Derivation cohort (n=226)

(B) Validation cohort (n=179)

Since this was beyond the scope of the current manuscript, we refrained from including this information in the manuscript, but would of course be ready to do so if requested by the Editors.

COMMENT #4: Could any of the biomarkers have worked better in combination with clinical features at admission, given that the bulk of the differentiation of deaths appears to be early?

RESPONSE #4: Thank you for this comment. Please refer to Response #2 to this reviewer's comment above, where we show that sTREM-1 significantly improves a validated clinical score from presentation. Our aim was to identify circulating biologic markers that can be

quantified without the need for a trained healthcare professional in a resource limited setting. Although we compared the use of a validated clinical severity of illness score (LODS), this was done to illustrate the excellent performance of a circulating biomarker with a biologic link to disease that can be incorporated into a point of care test to ease patient triage in the community.

Reviewer #4:

Major comments:

1. The study cohort was divided into a derivation cohort and an external validation cohort based on enrollment date. Such a strategy is crucial for developing and validating a data-driven predictor or classifier, where overfitting can be a serious issue and results can be overly optimistic without independent validation.

We thank the reviewer for acknowledging the challenges encountered when collecting data in resource limited settings and our best effort to overcome them by analyzing data in a derivation and external validation cohort based on enrollment date. The reviewer correctly points out that our aim was to prevent overfitting that can result in overly optimistic results using other approaches to select derivation and validation cohorts.

COMMENT #1: For this particular study, I would appreciate some clarifications on what exactly was derived from the derivation cohort and subsequently validated in the validation cohort. It seems that the 11 biomarkers under investigation were pre-defined (right?), and I guess the point of derivation/validation was to identify and validate the best biomarker with the largest AUC, which turned out to be sTREM-1? Was this planned prospectively?

RESPONSE #1: Thank you for complimenting the rationale for method selection for our study. With respect to the reviewer's question regarding what was derived in the derivation cohort and subsequently in the validation cohort: this included ascertaining the best circulating marker that predicted 7-day mortality, the cut-off point for the risk zones, and performance of the top markers in children with and without the diagnosis of malaria. The reviewer correctly points out that the biomarkers we studied were pre-defined as well as the fact that the point of the derivation/validation approach was to (1) identify and then (2) validate the biomarker with the largest AUC (sTREM-1). This was planned prospectively, as outlined in our pre-defined statistical analysis plan.

COMMENT #2: In Table 2 and elsewhere, you have presented results of an “internal validation” analysis. What exactly do you mean by “internal validation”? Does this refer to a cross-validation procedure in which a prediction is made for each subject based on independent data from other subjects in a training set? However, the ROC analysis for a given biomarker does not require any training at all. So I don't see the point of internal validation, and I don't understand why the results are not identical (I see they are very similar) for derivation and internal validation in Table 2 and Supplementary Tables 7-9. Please provide a rationale for the internal validation or remove it from the paper.

RESPONSE #2: As pointed out in the Methods Statistical analysis section, the internal validation was done as follows (page 16, paragraph 1):

“Discrimination based on AUROCs, associated rankings of biomarkers, likelihood ratios associated with identified cutoffs of sTREM-1 in the derivation cohort, calibration of mortality risks in green, yellow and red zones, and calibration of mortality risks predicted from log sTREM-1 levels were internally validated in the derivation cohort

based on 500 bootstrap samples with replacement, and externally validated in the validation cohort.”

As is usually the case, internal validation addresses overfitting – internal validation in combination with external validation adds considerably to the credibility of our results. As such, we would argue against removing the internal validation.

COMMENT #3: The Kaplan-Meier curves in Figure 3 are quite informative as they provide estimates of mortality rates that appropriately account for loss to follow-up. However, I find the associated text (L188-197) less informative for several reasons. First, “8.32 times more likely to die” is not an accurate interpretation of a hazard ratio (HR) of 8.32. Second, any HR is based on a proportional hazards assumption that is almost certainly false. Third, when correctly interpreted, the HR is just not very meaningful to non-statisticians. Why not just provide and compare 7-day mortality rates with confidence intervals in the text? The text can also include p-values for comparing different subgroups. If you really want to show HRs, they can be kept in Figure 3.

RESPONSE #3: As suggested, we now report the cumulative incidence at 7 days for each of the three zones with 95% confidence intervals and an omnibus p-value for differences in cumulative incidence between zones in the main text, and we removed HRs from Figure 3. We also report omnibus p-values for differences between zones in Figure 3 and supplementary Figures 5 and 6. All omnibus p-values were statistically significant.

The main results section has been modified (Page 7-8) as follows:

Fig. 3 presents time-to-event analyses in the overall population (top), in the subgroup of children with diagnosis of *P. falciparum* malaria (middle) and without diagnosis of malaria (bottom). In the derivation and validation cohorts combined, children in the green zone had an estimated incidence of death of 0.5% (95% CI 0.1 to 0.9%), children in the yellow zone 3.9% (95% CI 2.6 to 5.2%), and children in the red zone 31.8% (95% CI 25.4 to 38.2%; omnibus p-value for difference between 3 zones <0.0001). In children with diagnosis of *P. falciparum* malaria, those in the green zone had an estimated incidence of death of 0.0% (95% CI 0.0 to 0.5%), children in the yellow zone 1.6% (95% CI 0.5 to 2.8%), and children in the red zone 23.6% (95% CI 16.3 to 30.9%; omnibus p-value for difference between 3 zones 0.003). In children without diagnosis of *P. falciparum* malaria, those in the green zone had an estimated incidence of death of 0.9% (95% CI 0.1 to 1.6%), children in the yellow zone 7.4% (95% CI 4.7 to 10.2%), and children in the red zone 45.9% (95% CI 34.5 to 57.3%; omnibus p-value for difference between 3 zones <0.0001). **Supplementary Fig. 5** and **6** show corresponding time-to-event analyses in derivation (**Supplementary Fig. 5**) and validation (**Supplementary Fig. 6**) cohorts separately, which showed similar results.

COMMENT #4: For children with malaria, you have indicated that “Cox models were unstable” and you therefore used Poisson regression to estimate rate ratios. What was the issue with Cox models? Not enough events? What was the rationale for using Poisson regression? Please provide a justification (e.g., references) for using Poisson regression in this setting, or else remove this analysis.

RESPONSE #4: In line with our response to comment #3, we now report cumulative incidence at 7 days for each zone with 95% confidence intervals in Figure 3 throughout. This avoids inconsistency in estimates between Panel B and the remaining Panels.

COMMENT #5: [Optional] Table 1 shows that a number of baseline characteristics (other than biomarkers) are predictive of vital status at 7 days. It is of interest to see how sTREM-1 (and other biomarkers) may add to those baseline characteristics for predicting vital status at 7 days. A simple way to look at this would be comparing predictions based on logistic regression models that include the variables in Table 1 with or without sTREM-1 (and possibly other biomarkers).

RESPONSE #5: Please see our response to Comment #2 of Reviewer #3. Briefly, our goal was to identify circulating markers that could be quantified at the point of care without the requirement of a trained healthcare professional to ascertain clinical signs and symptoms. As such, the focus of our analysis was the comparative performance that focused only on the biological markers with a pathophysiological link to sepsis.

Of note, the addition of clinical signs/symptoms to a univariate model that included only sTREM-1 did not improve predictive capacity.

Minor comments:

COMMENT #6: L128-131: “An additional 376 children were excluded...as they had absconded or were transferred before or after 7 days (Fig. 1).” I understand that their vital status at 7 days could not be ascertained if they were lost to follow-up before 7 days. I don’t understand what the problem is with those children who were lost to follow-up AFTER 7 days. Don’t we know already that they were alive at 7 days?

RESPONSE #6: As indicated in our response to comments #3 and #6 of Reviewer #2, children who absconded, were transferred, or for whom vital status was unknown were only excluded from the analysis of the comparative performance of 11 biomarkers, as biomarkers originally were not measured in this subgroup, following a historical decision of the investigators made in 2016. This decision was made at the time even though the vital status up to 7 days was known in those children who absconded or were transferred later than 7 days.

However, sTREM-1 and sFlt-1, which ranked first and second in the analysis of the derivation cohort, sTNFR1, which ranked second after sTREM-1 in the previously reported study in febrile adults in Tanzania, and Ang-2, which was the biomarker used for sample size considerations of the current study, were subsequently quantified using identical methods in the remaining 376 children with available plasma who were transferred or absconded.

The main analysis on the prognostic performance of sTREM-1 is therefore based on all 2460 children with available plasma, including the 376 children referred to by this reviewer.

COMMENT #7: L183-186: Could you explain somewhere how the calibration was done for Supplementary Figures 2 and 3? This is not trivial because some subjects were lost to follow-up.

RESPONSE #7: Calibration plots were created after multiple imputation to account for missing data, using 20 imputed datasets. We plotted observed against expected probabilities for assessment of prediction model performance.

We used the STATA command `pmcalplot`, which produces a calibration plot of observed against expected probabilities for assessment of prediction model performance. Calibration is plotted in groups across the risk spectrum as recommended in the TRIPOD guidelines. The spike plot of the distribution of events and non-events was displayed using observed events to ensure appropriate representation of the data.

COMMENT #8: L326-327: The AUROC can be, and usually is, estimated nonparametrically. Why do you use logistic regression to estimate the AUROC?

RESPONSE #8: Thank you for spotting this error. We used the command `roccomp` in STATA, which indeed is based on a non-parametric method originally described by DeLong et al.²³ We amended the methods section accordingly and cite the paper by DeLong et al.

COMMENT #9: L359-360: "...internally validated...based on 500 bootstrap samples..." This doesn't make sense to me. Bootstrap is a general approach to variance estimation and inference. I still don't know what the internal validation was.

RESPONSE #9: We cite Steyerberg's book on clinical prediction models (Steyerberg E. *Clinical Prediction Models: A Practical Approach to Development, Validation, and Updating*. New York: Springer-Verlag; 2009)³⁵ when describing bootstrap validation. Please refer to Chapter 17, which describes the approach as follows:

"As discussed in Chap. 5, bootstrapping reflects the process of sampling from the underlying population. Bootstrap samples are drawn with replacement from the original sample, reflecting the drawing of samples from an underlying population. Bootstrap samples are of the same size as the original sample. In the context of model validation, 100–200 bootstraps may often be sufficient to obtain stable estimates, but in one simulation study we reached a plateau only after 500 bootstrap repetitions. With current computer power bootstrap validation is a feasible technique for most prediction problems.

For bootstrap validation a prediction model is developed in each bootstrap sample. This model is evaluated both in the bootstrap sample and in the original sample. The first reflects apparent validation, the second test validation in new subjects. The difference in performance indicates the optimism. This optimism is subtracted from the apparent performance of the original model in the original sample. The bootstrap was illustrated for estimation of optimism in Chap. 5.

Advantages of bootstrap validation are various. The optimism-corrected performance estimate is rather stable, since samples of size N are used to develop the model as well as to test the model. This is similar to apparent validation, and an advantage over split-sample and cross-validation methods. Compared with apparent validation, some uncertainty is added by having to estimate the optimism. When sufficient bootstraps are taken, this additional uncertainty is however negligible.

Moreover, simulations have shown that bootstrap validation can appropriately reflect all sources of model uncertainty, especially variable selection.”

COMMENT #10: L376-379: A multiple imputation approach was used to deal with missing data on vital status at 7 days, “with all baseline characteristics and in-hospital death up to 7 days as variables in the imputation model to create 20 imputed datasets.” I presume that “in-hospital death up to 7 days” was treated as the response variable, right? Could you specify the baseline variables used in the imputation model? Are they the same ones listed in Table 1?

RESPONSE #10: The multiple imputation model allows to use both, baseline characteristics and outcomes as covariates without distinction. Baseline characteristics and vital status at 7 days were used as predictors, but were also imputed when missing, irrespective of being a baseline characteristic or an outcome. Note that biomarkers were not imputed, as described in the methods section.

The variables used in the multiple imputation model are now specified in the supplementary information, and included: hospital death, duration of follow up, abscondment, age, sex, LODS score, temperature, oxygen saturation, heart rate, lactate, time to evaluation by medical doctor, malaria status, HIV status, pneumonia diagnosis, log-transformed biomarker values (sTREM-1, Ang2, sFlt-1, sTNFR1)

COMMENT #11: Table 1: Are these results based on univariate analyses (logistic regression of vital status on one baseline variable at a time)?

RESPONSE #11: Yes, indeed. We clarified this in the legend as follows:
“Odds ratios, 95% confidence intervals and p-values are from univariate analyses.”

REFERECES:

1. Goldstein, B., Giroir, B., Randolph, A. & International Consensus Conference on Pediatric, S. International pediatric sepsis consensus conference: definitions for sepsis and organ dysfunction in pediatrics. *Pediatr Crit Care Med* **6**, 2-8 (2005).
2. Rudd, K.E., *et al.* Global, regional, and national sepsis incidence and mortality, 1990-2017: analysis for the Global Burden of Disease Study. *Lancet* **395**, 200-211 (2020).
3. Dondorp, A.M., *et al.* Artesunate versus quinine in the treatment of severe falciparum malaria in African children (AQUAMAT): an open-label, randomised trial. *Lancet* **376**, 1647-1657 (2010).
4. Jaeschke, R., Guyatt, G.H. & Sackett, D.L. Users' guides to the medical literature. III. How to use an article about a diagnostic test. B. What are the results and will they help me in caring for my patients? The Evidence-Based Medicine Working Group. *JAMA* **271**, 703-707 (1994).
5. Furukawa, T.A., Strauss, S.E., Bucher, H.C., Thomas, A. & Guyatt, G. Diagnostic Tests. in *Users' Guides to the Medical Literature: A Manual for Evidence-Based Clinical Practice* (eds. Gordon Guyatt, Drummond Rennie, Maureen O. Meade & Cook., D.J.) 419-438 (McGraw-Hill, New York, 2014).
6. Leligdowicz, A., Richard-Greenblatt, M., Wright, J., Crowley, V.M. & Kain, K.C. Endothelial Activation: The Ang/Tie Axis in Sepsis. *Front Immunol* **9**, 838 (2018).
7. Xing, K., Murthy, S., Liles, W.C. & Singh, J.M. Clinical utility of biomarkers of endothelial activation in sepsis--a systematic review. *Crit Care* **16**, R7 (2012).
8. Erdman, L.K., *et al.* Combinations of host biomarkers predict mortality among Ugandan children with severe malaria: a retrospective case-control study. *PloS one* **6**, e17440 (2011).
9. Erdman, L.K., *et al.* Chitinase 3-like 1 is induced by Plasmodium falciparum malaria and predicts outcome of cerebral malaria and severe malarial anaemia in a case-control study of African children. *Malar J* **13**, 279 (2014).
10. Mikacenic, C., *et al.* Biomarkers of Endothelial Activation Are Associated with Poor Outcome in Critical Illness. *PloS one* **10**, e0141251 (2015).
11. Hack, C.E., *et al.* Increased plasma levels of interleukin-6 in sepsis. *Blood* **74**, 1704-1710 (1989).
12. Marty, C., *et al.* Circulating interleukin-8 concentrations in patients with multiple organ failure of septic and nonseptic origin. *Crit Care Med* **22**, 673-679 (1994).
13. Ng, P.C., *et al.* IP-10 is an early diagnostic marker for identification of late-onset bacterial infection in preterm infants. *Pediatr Res* **61**, 93-98 (2007).
14. Ricciuto, D.R., *et al.* Angiopoietin-1 and angiopoietin-2 as clinically informative prognostic biomarkers of morbidity and mortality in severe sepsis. *Crit Care Med* **39**, 702-710 (2011).
15. Poukoulidou, T., *et al.* TREM-1 expression on neutrophils and monocytes of septic patients: relation to the underlying infection and the implicated pathogen. *BMC infectious diseases* **11**, 309 (2011).
16. Emmerich, C.H., *et al.* Improving target assessment in biomedical research: the GOT-IT recommendations. *Nature Reviews Drug Discovery* (2020).
17. Ghosh, S., Baranwal, A.K., Bhatia, P. & Nallasamy, K. Suspecting Hyperferritinemic Sepsis in Iron-Deficient Population: Do We Need a Lower Plasma Ferritin Threshold? *Pediatr Crit Care Med* **19**, e367-e373 (2018).
18. Leligdowicz, A., *et al.* Validation of two multiplex platforms to quantify circulating markers of inflammation and endothelial injury in severe infection. *PloS one* **12**, e0175130 (2017).

19. Richard-Greenblatt, M., *et al.* Prognostic Accuracy of Soluble Triggering Receptor Expressed on Myeloid Cells (sTREM-1)-based Algorithms in Febrile Adults Presenting to Tanzanian Outpatient Clinics. *Clin Infect Dis* **70**, 1304-1312 (2020).
20. Sterne, J.A., *et al.* Multiple imputation for missing data in epidemiological and clinical research: potential and pitfalls. *BMJ* **338**, b2393 (2009).
21. Conroy, A.L., *et al.* Prospective validation of pediatric disease severity scores to predict mortality in Ugandan children presenting with malaria and non-malaria febrile illness. *Crit Care* **19**, 47 (2015).
22. WHO. Handbook : IMCI integrated management of childhood illness. (Geneva, Switzerland, 2005).
23. DeLong, E.R., DeLong, D.M. & Clarke-Pearson, D.L. Comparing the areas under two or more correlated receiver operating characteristic curves: a nonparametric approach. *Biometrics* **44**, 837-845 (1988).
24. Higgins, S.J., *et al.* Dysregulation of angiopoietin-1 plays a mechanistic role in the pathogenesis of cerebral malaria. *Science translational medicine* **8**, 358ra128 (2016).
25. Wong, H.R., *et al.* The pediatric sepsis biomarker risk model. *Crit Care* **16**, R174 (2012).
26. Wong, H.R., *et al.* Testing the prognostic accuracy of the updated pediatric sepsis biomarker risk model. *PloS one* **9**, e86242 (2014).
27. Wong, H.R., *et al.* A multibiomarker-based outcome risk stratification model for adult septic shock*. *Crit Care Med* **42**, 781-789 (2014).
28. Wong, H.R., *et al.* Pediatric Sepsis Biomarker Risk Model-II: Redefining the Pediatric Sepsis Biomarker Risk Model With Septic Shock Phenotype. *Crit Care Med* **44**, 2010-2017 (2016).
29. Wong, H.R., *et al.* Improved Risk Stratification in Pediatric Septic Shock Using Both Protein and mRNA Biomarkers. PERSEVERE-XP. *Am J Respir Crit Care Med* **196**, 494-501 (2017).
30. Wong, H.R., *et al.* Prospective clinical testing and experimental validation of the Pediatric Sepsis Biomarker Risk Model. *Science translational medicine* **11**(2019).
31. Conroy, A.L., *et al.* Host Biomarkers Are Associated With Response to Therapy and Long-Term Mortality in Pediatric Severe Malaria. *Open Forum Infect Dis* **3**, ofw134 (2016).
32. Helbok, R., *et al.* The Lambarene Organ Dysfunction Score (LODS) is a simple clinical predictor of fatal malaria in African children. *J Infect Dis* **200**, 1834-1841 (2009).
33. Berkley, J.A., *et al.* Prognostic indicators of early and late death in children admitted to district hospital in Kenya: cohort study. *BMJ* **326**, 361 (2003).
34. Gupta, M.A., *et al.* Validation of "Signs of Inflammation in Children that Kill" (SICK) score for immediate non-invasive assessment of severity of illness. *Ital J Pediatr* **36**, 35 (2010).
35. Steyerberg, E.W. *Clinical Prediction Models: A Practical Approach to Development, Validation, and Updating*, (Springer, New York, NY, 2009).

REVIEWERS' COMMENTS

Reviewer #1 (Remarks to the Author):

The manuscript is improved. However several points of improvement remain.

- Measurements of ferritin and PCT should be done in stored samples and reported.
- Reference to publications of the 2000s on the diagnostic performance of sTREM-1 in adult sepsis is missing.

Reviewer #2 (Remarks to the Author):

Thank you for considering some of the recommendations of the prior reviews. It is appreciated state that the authors have prespecified their analysis plan. However, there is a philosophical mismatch between the way the manuscript is written and what the methods and data reflect. This may be my perception, but I think there is an important distinction to be made. The manuscript reads as though it is fitting into a 'risk prediction model' framework. However, there is no model fitting and the idea of derivation and validation is misleading.

My interpretation of the study is that a selected group of 11 biomarkers with hypothesized mechanistic association with the outcome of interest were chosen for evaluation based on availability. Among a cohort of patients at risk, the diagnostic test characteristics (AUC) were estimated. This was used to select from among the biomarkers those to carry forward to (I think) a third cohort for comprehensive evaluation among all persons, including those previously excluded because of practical and cost considerations (i.e. an unbiased cohort). Diagnostic accuracy was repeated in each evaluation, with sTREM1 outperforming other biomarkers. The cohorts were subsequently combined and global diagnostic test statistics reported.

Perhaps I am confused by the idea of derivation in the absence of modeling, but I find the very simple and important message of the paper I outlined above is diluted with the multiple cohorts and evaluations described as derivations and validations. At its root, this manuscript demonstrates clearly that a single, easy to measure biomarker has good decision-supporting (diagnostic) accuracy

that could help in resource poor settings were the similarly performing clinical alternative, the LODS, would not be feasible.

I recognize this might be just an alternative way of describing what are robust and clear data supporting the potential utility of sTREM1 in remote settings, but it should be clear that this study is not attempting to derive and validate risk prediction models.

Reviewer #3 (Remarks to the Author):

I believe the authors have extensively and thoughtfully responded to the critique and questions. I would suggest that as offered, the specific details in response to Reviewer 3 (comments 2 and 3) be included in the Supplementary materials

Reviewer #4 (Remarks to the Author):

All of my previous comments have been addressed. I have no further comments.

RESPONSES TO REVIEWER COMMENTS

Reviewer #1

The manuscript is improved. However several points of improvement remain.

- Measurements of ferritin and PCT should be done in stored samples and reported.

Reviewer 1, Response #1:

Based on the Editor's request, we did not include ferritin analyses in our final revised manuscript and instead discussed limitations of their measurement in the discussion section.

“Lastly, other acute phase proteins, such as ferritin, are increased during infection. Although this marker is associated with severity of illness, it is not specific to the pathobiology of febrile illness and as such, was not included in this study. Of note, the predictive performance of 7-day mortality of CRP and PCT, well-studied non-specific markers of inflammation, was significantly lower for relative to sTREM-1.”

We included PCT analyses in the results section (as also requested by Reviewer 3):

- Reference to publications of the 2000s on the diagnostic performance of sTREM-1 in adult sepsis is missing.

Reviewer 1, Response #2:

Recent references to manuscripts on the diagnostic performance of sTREM-1 in adults sepsis were added to the discussion section of the manuscript. These included:

- *Wright, S.W., et al. sTREM-1 predicts mortality in hospitalized patients with infection in a tropical, middle-income country. BMC Med 18, 159 (2020).*
- *Chang, W., Peng, F., Meng, S.-S., Xu, J.-Y. & Yang, Y. Diagnostic value of serum soluble triggering expressed receptor on myeloid cells 1 (sTREM-1) in suspected sepsis: a meta-analysis. BMC Immunology 21, 2 (2020).*

Reviewer #2:

Thank you for considering some of the recommendations of the prior reviews. It is appreciated state that the authors have prespecified their analysis plan. However, there is a philosophical mismatch between the way the manuscript is written and what the methods and data reflect. This may be my perception, but I think there is an important distinction to be made. The manuscript reads as though it is fitting into a 'risk prediction model' framework. However, there is no model fitting and the idea of derivation and validation is misleading.

My interpretation of the study is that a selected group of 11 biomarkers with hypothesized mechanistic association with the outcome of interest were chosen for evaluation based on availability. Among a cohort of patients at risk, the diagnostic test characteristics (AUC) were

estimated. This was used to select from among the biomarkers those to carry forward to (I think) a third cohort for comprehensive evaluation among all persons, including those previously excluded because of practical and cost considerations (i.e. an unbiased cohort). Diagnostic accuracy was repeated in each evaluation, with sTREM1 outperforming other biomarkers. The cohorts were subsequently combined and global diagnostic test statistics reported.

Perhaps I am confused by the idea of derivation in the absence of modeling, but I find the very simple and important message of the paper I outlined above is diluted with the multiple cohorts and evaluations described as derivations and validations. At its root, this manuscript demonstrates clearly that a single, easy to measure biomarker has good decision-supporting (diagnostic) accuracy that could help in resource poor settings were the similarly performing clinical alternative, the LODS, would not be feasible.

I recognize this might be just an alternative way of describing what are robust and clear data supporting the potential utility of sTREM1 in remote settings, but it should be clear that this study is not attempting to derive and validate risk prediction models.

Reviewer 2, Response #1:

Thank you for the thorough review of the complexity of our analysis. We agree that the intension of our manuscript is to provide the simple message that a single biomarker has the capacity to risk-stratify patients at the time of clinical presentation.

For logistic reasons (time, cost), we were able to only conduct our study at a single clinical site. To emulate the setting in which a derivation cohort and validation cohort was feasible, we used time to dichotomize our single cohort into two cohorts to ensure that the biomarker we identified as the most predictive of death in febrile children was robust. Indeed, it is true we are not deriving and validating risk prediction models but rather providing evidence that in two different sets of patients separated by time, sTREM-1 has the greatest discrimination, and as such, is a very promising biomarker for future clinical studies to enable risk stratification of febrile children at risk of death in a resource limited setting.

Reviewer #3:

I believe the authors have extensively and thoughtfully responded to the critique and questions. I would suggest that as offered, the specific details in response to Reviewer 3 (comments 2 and 3) be included in the Supplementary materials

Reviewer 3, Response #1:

The information included in our initial reply to Reviewer 3, comments 2 and 3, are now mentioned in the result section of the main manuscript as follows:

(A) Predictive performance of sTREM-1 relative to clinical scoring systems

The predictive performance of sTREM-1 relative to the best clinical score (LODS) was statistically similar in the derivation (AUROC for sTREM 0.894 (95%-CI 0.843 to 0.944) versus

for LODS 0.907 (95%-CI 0.869 to 0.944), $P = 0.661$) and in the validation (AUROC for sTREM 0.901 (95%-CI 0.856 to 0.946) versus for LODS 0.912 (95%-CI 0.875 to 0.949; $P = 0.628$) cohort.

(B) Predictive performance of sTREM-1 relative to PCT and CRP

In contrast, sTREM-1 predictive performance of 7-day mortality relative to C-reactive protein (CRP) and procalcitonin (PCT) was significantly greater in the derivation (AUROC for sTREM-1 0.872 (95%-CI 0.812 to 0.932) versus for CRP 0.645 (95%-CI 0.547 to 0.742), $P < 0.0001$ and for PCT 0.667 (95%-CI 0.565 to 0.769), $P < 0.0001$) and in the validation (AUROC for sTREM-1 0.879 (95%-CI 0.823 to 0.936) versus for CRP 0.597 (95%-CI 0.501 to 0.693), $P < 0.0001$ and for PCT 0.695 (95%-CI 0.605 to 0.785), $P < 0.0001$) cohort.

Reviewer #4:

All of my previous comments have been addressed. I have no further comments.